# Cartan Networks: Group theoretical Hyperbolic Deep Learning

## Abstract

Hyperbolic deep learning leverages the metric properties of hyperbolic spaces to develop efficient and informative embeddings of hierarchical data. Here, we focus on the solvable group structure of hyperbolic spaces, which follows naturally from their construction as symmetric spaces. This dual nature of Lie groups and Riemannian manifolds allows us to propose a new class of hyperbolic deep learning algorithms where group homomorphisms are interleaved with metric-preserving diffeomorphisms. The resulting algorithms, which we call *Cartan networks*, show promising results on various benchmark datasets and open the way for a novel class of hyperbolic deep learning architectures.

## 1 Introduction

The concept of distance is the core of machine learning and pattern recognition. While much classical machine learning can be recast as learning distances directly from data (e.g. Bishop (2006)), recent developments have pointed out that common data structures, such as trees and graphs, cannot be easily accommodated within Euclidean spaces, thus requiring a more radical rethink of the geometry of data spaces. In this context, the $n$-dimensional hyperbolic space $\mathbb{H}^n$ has received significant attention as a suitable space in which to embed hierarchically structured data (Nickel & Kiela, 2017), spurring a productive line of research combining hyperbolic geometry with various deep learning architectures (Ganea et al., 2018; Chami et al., 2019; Gulcehre et al., 2019; Shimizu et al., 2021; Chen et al., 2022; Peng et al., 2022; Bdeir et al., 2024). These so-called *hyperbolic neural networks* have found applications in fields as diverse as neuroscience (Gao et al., 2020), single-cell transcriptomics (Klimovskaia et al., 2020), and recommender systems (Chamberlain et al., 2019).

Geometrically $\mathbb{H}^n$ is a $n$-dimensional hyperboloid, namely the quadric locus $\sum_{i=1}^{n} X_i^2 - X_{n+1}^2 = -1$ in $\mathbb{R}^{n+1}$. It is also a *coset manifold*, namely the quotient of a Lie Group modulo a maximal Lie subgroup, $\mathbb{H}^n \simeq \mathrm{SO}(1,n)/\mathrm{SO}(n)$, and more specifically a *symmetric space*. The study and classification of symmetric spaces is one of the monumental achievements of the French mathematician Èlie Cartan (Cartan, 1926; Helgason, 1962; Magnea, 2002; Fré, 2023). The non-compact symmetric spaces are all metrically equivalent to a corresponding solvable Lie group $S$ of the same dimension, a mathematical result that was discovered and developed in the context of Supergravity Theory (Andrianopoli et al., 1997b;a; Fré et al., 2007; Alekseevsky, 1975; Cortés, 1996; Alekseevsky et al., 2004), and amply reviewed and systematically reorganized for machine learning applications in Bruzzo et al. (2025).

This result, to our knowledge, is not known so far in the machine learning literature, and has significant algorithmic consequences. The dual nature of group and Riemannian manifold of the hyperbolic space $\mathbb{H}^n$ enables us to construct a deep learning framework based entirely on *intrinsic* geometric operations, where group homomorphisms are interleaved with metric-preserving diffeomorphisms in creating a powerful function approximation machine. Importantly, the nonlinearities naturally arising from group-theoretic exponential and logarithmic maps give flexibility to the framework, which achieves promising results on benchmark datasets when compared with similar-sized standard deep learning architectures.

The main contributions of this work are as follows:

- We highlight the metric equivalence of the hyperbolic space with a solvable Lie group to exploit the group structure as a tool in architectural design.

- We propose a new deep learning architecture where each layer is a solvable Lie group $S_i$ and where the map from layer $i$ to layer $i+1$ can be represented as a combination of homomorphisms from the solvable Lie group $S_i$ to the next one $S_{i+1}$ and the isometries of $S_{i+1}$. The construction is general for any symmetric space, and we implement it for the hyperbolic space $\mathbb{H}^n$.
- We extensively benchmark these architectures on real and synthetic datasets, showing competitive or better performance w.r.t. Euclidean and standard hyperbolic neural networks.

### 1.1 PREVIOUS LITERATURE

Early works in hyperbolic deep learning focused on hyperbolic embeddings for hierarchical data. (Nickel & Kiela, 2017) introduced *Poincaré embeddings*, showing superior hierarchical representation compared to Euclidean embeddings. Ganea et al. (2018) and subsequent works (Shimizu et al., 2021; Chen et al., 2022; Bdeir et al., 2024; Peng et al., 2022), extended hyperbolic geometry to deep learning by developing *hyperbolic neural networks*, using Möbius operations (Ungar, 2009). Various generalizations of hyperbolic networks have been explored. Convolutional networks (Dai et al., 2021; Skliar & Weiler, 2023; Ghosh et al., 2024), graph neural networks (Chami et al., 2019), and attention mechanisms (Gulcehre et al., 2019) hyperbolic variants were introduced to handle different datasets, as well as methods of dimensionality reduction (Chami et al., 2021; Fan et al., 2022).

Lie groups and Lie algebras are often studied in deep learning for their equivariance properties (Cohen et al., 2019; Chen et al., 2020; Otto et al., 2024). Architectures based on semisimple Lie algebras have been introduced under the name Lie Neurons (Lin et al., 2024), focusing on making these networks adjoint-equivariant.

The notion that $\mathbb{H}^n$ is isometric to a Lie group was explored in the context of probability distributions and Frechét means by Jaćimović (2025). However, the isometry between symmetric spaces and solvable groups was not highlighted in full generality, and the knowledge was never applied to the study of deep learning architectures.

## 2 THEORETICAL PRELIMINARIES

We will assume basic knowledge of Lie groups (see Appendix A for a brief introduction).

**Solvable groups and Cartan subalgebras.**

**Definition 2.1** (Subalgebra commutator). Let $\mathfrak{h}_1, \mathfrak{h}_2$ be two subalgebras of $\mathfrak{g}$. Their commutator subalgebra is

$$[\mathfrak{h}_1, \mathfrak{h}_2] := \{[h_1, h_2] \in \mathfrak{g} \mid h_1 \in \mathfrak{h}_1, \ h_2 \in \mathfrak{h}_2\} \tag{1}$$

where $[\,\cdot\,,\,\cdot\,]$ denotes the Lie bracket of the algebra.

**Definition 2.2** (Derived series). Let $\mathfrak{g}$ be a Lie algebra. Its *derived series* is the series

$$\mathfrak{g}^{(0)} = \mathfrak{g}, \qquad \mathfrak{g}^{(n+1)} = [\mathfrak{g}^{(n)}, \mathfrak{g}^{(n)}] \quad \forall n \in \mathbb{N} \tag{2}$$

The derived series is a decreasing sequence of ideals in the algebra.

**Definition 2.3** (Solvable algebras). A Lie algebra $\mathfrak{g}$ is *solvable* if its derived series is eventually 0, that is to say, if

$$\exists n \in \mathbb{N} \quad \text{s.t.} \quad \mathfrak{g}^{(n)} = 0$$

A Lie group is solvable if its Lie Algebra is solvable.

In practice, solvable groups are best understood in terms of their matrix representation. In fact,

**Theorem 2.1** (Lie's theorem (Humphreys, 1972)). *Let $\mathfrak{g}$ be a solvable subalgebra of the general linear group $\mathfrak{gl}_V$. Then there exists a basis of $V$ with respect to which $\mathfrak{g}$ is made of upper triangular matrices.*

This theorem shows we can think of solvable groups as upper-triangular matrix Lie groups.

**Definition 2.4** (Cartan subalgebras). Let $\mathfrak{g} \subseteq \mathfrak{gl}_n(\mathbb{R})$ be a matrix Lie algebra consisting of upper triangular matrices. Its *Cartan subalgebra* is the subspace of diagonal matrices.

**Symmetric spaces.** Let $G$ be a Lie group and $H$ a normal subgroup, $\mathfrak{g}$ and $\mathfrak{h}$ the corresponding Lie algebras. A coset manifold G/H is a symmetric space if and only if there is an orthogonal decomposition of $\mathfrak{g}$, as a vector space, as follows:

$$\mathfrak{g} = \mathfrak{h} \oplus \mathfrak{m} \quad ; \quad \begin{cases} [\mathfrak{h}, \mathfrak{h}] & \subset & \mathfrak{h} \\ [\mathfrak{h}, \mathfrak{m}] & \subset & \mathfrak{m} \\ [\mathfrak{m}, \mathfrak{m}] & \subset & \mathfrak{h} \end{cases} \tag{3}$$

One interesting class of non-compact symmetric spaces is given by

$$\mathcal{M}^{[r,r+p]} = \frac{\mathrm{SO}(r, r+p)}{\mathrm{SO}(r) \times \mathrm{SO}(r+p)}, \ r > 0, \ p \geq 0 \tag{4}$$

This family of manifolds is easily tractable thanks to the metric equivalence between these and an appropriate *solvable Lie group*, studied in the context of theoretical physics in Bruzzo et al. (2025),

$$\mathcal{M}^{[r,p]} \simeq \mathrm{Exp}\left[\mathrm{Solv}_{[r,p]}\right]$$

where we denote $\mathrm{Solv}_{[r,p]}$ the solvable Lie algebra of the solvable Lie subgroup $S_{[r,p]} \subset \mathrm{SO}(r, r+p)$ with $r$ Cartan generators. For $r = 1$ we realize the hyperbolic space $\mathbb{H}^{p+1} \simeq \mathcal{M}^{[1, 1+p]}$ (where $\simeq$ denotes a metric equivalence).

**Solvable coordinates of hyperbolic space.** The hyperbolic space $\mathbb{H}^n$ (and all the other non-compact symmetric spaces) is metrically equivalent to an appropriate solvable Lie group, whose structure was never used in statistical learning.

$$\mathbb{H}^{q+1} \simeq \frac{\mathrm{SO}(1, 1+q)}{\mathrm{SO}(1+q)} = \mathcal{M}^{[1, 1+q]} \simeq \mathrm{Exp}\left[\mathrm{Solv}_{[1,1+q]}\right] \tag{5}$$

As this manifold is a Lie group, we will parametrize the manifold with a set of coordinates

$$\Upsilon = [\Upsilon_1, \boldsymbol{\Upsilon_2}]^\intercal = [\Upsilon_1, \Upsilon_{2,1}, \ldots, \Upsilon_{2,q}]^\intercal, \tag{6}$$

called the *solvable coordinates* of the manifold (Bruzzo et al., 2025), and we will use them for our formulation of hyperbolic learning. We separate the first component $\Upsilon_1$ (which we will call the Cartan coordinate since it corresponds to the unique generator of the Cartan subalgebra) from the others (which we call the paint coordinates following Bruzzo et al. (2025). This choice of coordinate system for the hyperbolic space is convenient for many reasons discussed throughout this work. A convenient property of all non-compact symmetric spaces is that they can be easily parametrized by a single chart with domain $\mathbb{R}^n$, thus bypassing the numerical problems of the Lorentz and Poincaré models exposed by Mishne et al. (2023).

**Group operation.** The group operation is the matrix multiplication between the solvable group elements. Given two points $\Upsilon, \Psi \in \mathcal{M}^{[1, q]}$, the group operation is

$$\Psi * \Upsilon = \begin{bmatrix} \Upsilon_1 + \Psi_1 \\ \boldsymbol{\Upsilon_2} + \mathrm{e}^{-\Upsilon_1} \boldsymbol{\Psi_2} \end{bmatrix} \tag{7}$$

Similarly, the inverse element is given by $\Upsilon^{-1} = \left[-\Upsilon_1, -\mathrm{e}^{\Upsilon_1} \boldsymbol{\Upsilon_2}\right]^\intercal$. The matrix representative is expressed in Eq. 23 in Appendix B, alongside a deeper discussion of the solvable coordinates parametrization, and the identity element is the point $\Upsilon = \mathbf{0}$. The group operations can be expressed in terms of the non-solvable Poincaré ball coordinates (see Eq. 30 in Appendix B for the transition function) or other coordinate systems. Appendix C discusses various Riemannian operations in this coordinate system, including the distance between points.

## 3 LEARNING IN SYMMETRIC SPACES

### 3.1 GENERAL PRINCIPLES OF CARTAN NETWORKS

We propose creating a network whose layers are a sequence of solvable groups $\{S_i\}_{i=1}^N$.

The map from layer $i-1$ to layer $i$ is the composition of a group homomorphism with an isometry of the target space. Specifically, each transformation consists of a homomorphism (a map between groups that preserves the group operation):

$$h_i(W_i) \, : \, S_{i-1} \longrightarrow S_i, \tag{8}$$

from one solvable Lie group to the next, defined intrinsically by parameters $W_i$, composed with an isometry (a metric-preserving, and thus distance-preserving, map) acting on $S_i$:

$$\varphi_i(\theta_i) \, : \, S_i \longrightarrow S_i \tag{9}$$

parametrized by $\theta_i$. In the following, we develop the architecture in the case of the hyperbolic space, so $S_i \simeq \mathcal{M}^{[1,1+q_i]}$.

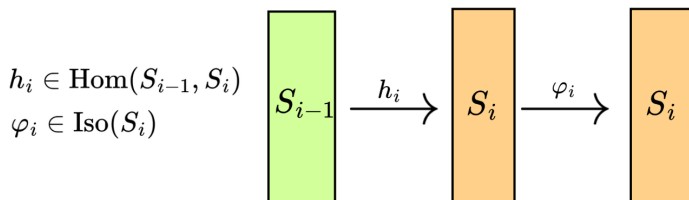

$$\mathbb{R}^d \to S_1 \to S_2 \to \cdots \to S_{i-1} \to S_i \to \cdots \to S_N \to [0,1]$$

Figure 1: **Structure of Cartan network (binary classification).** This figure illustrates the composition of the proposed Cartan networks between symmetric spaces. By alternating homomorphisms and isometries, our networks parametrize a larger class of maps while only using geometrically motivated functions.

### 3.2 MAPS BETWEEN HYPERBOLIC SPACES

**Isometries.** The set of isometries of $\mathcal{M}^{[1, 1+q]}$ into itself is given by $\mathrm{SO}(1, 1 + q)$ (these are parameterized in terms of the Poincarè ball coordinates by Jaćimović (2025); Jaćimović & Crnkić (2025). These isometries are a composition of three distinct isometries, namely the *paint rotation* (an orthogonal transformation of the paint coordinates $\Upsilon_2$), the *group operation*, and the *fiber rotation*, which mixes Cartan and paint coordinates. Of these, only the paint rotation is also a homomorphism of the group into itself. Refer to Appendix D for a detailed derivation.

A general isometry $\varphi \in \mathrm{Iso}(\mathcal{M}^{[1, 1+q]})$ can be parametrized as

$$\varphi(\Upsilon; Q, \beta, u) = R_{\boldsymbol{u}} \left( \begin{bmatrix} \beta_1 \\ \boldsymbol{\beta}_2 \end{bmatrix} * \begin{bmatrix} 1 & 0 \\ 0 & Q \end{bmatrix} \begin{bmatrix} \Upsilon_1 \\ \boldsymbol{\Upsilon}_2 \end{bmatrix} \right), \tag{10}$$

where $Q \in \mathrm{SO}(q)$, $\beta \in \mathcal{M}^{[1, 1+q]}$, $\boldsymbol{u} \in \mathbb{S}^{q+1}$ is a parameter on the n-sphere, and the fiber rotation $R_{\boldsymbol{u}}$ is given by

$$R_u(\Upsilon) = \begin{bmatrix} -\log\left(-\frac{1}{2}(e^{\Upsilon_1}(1+\|\boldsymbol{\Upsilon}_2\|^2) + e^{-\Upsilon_1})(1+u_0) + e^{-\Upsilon_1}u_0 - \boldsymbol{\Upsilon}_2 \cdot \boldsymbol{u}'\right) \\ \boldsymbol{\Upsilon}_2 - \left(\frac{\boldsymbol{\Upsilon}_2 \cdot \boldsymbol{u}'}{1+u_0} + \frac{1}{2}(e^{\Upsilon_1}(1+\|\boldsymbol{\Upsilon}_2\|^2) - e^{-\Upsilon_1})\right)\boldsymbol{u}' \end{bmatrix}, \quad (11)$$

having defined $\boldsymbol{u} = [u_0, u_1, \ldots, u_q]^\mathsf{T} \in \mathbb{S}^{q+1}$, and $\boldsymbol{u}' = [u_1, \ldots, u_q]^\mathsf{T}$.

**Solvable group homomorphisms.** The set of group homomorphisms is given by the linear maps between the corresponding solvable algebras that preserve the group structure. These are not linear in the coordinates in general, but the equations simplify in the $r = 1$ case. This class of transformations is the primary innovation of our architectures. It is important to note that, since the metric is left-invariant but not bi-invariant, the Riemannian logarithmic map and the Lie logarithmic map are not equivalent. If they were, our formulation would reduce to the same set of functions introduced by Ganea et al. (2018).

**Theorem 3.1.** *Let $h \in Hom(\mathcal{M}^{[1, 1+q]}, \mathcal{M}^{[1, 1+p]})$, $p, q \geq 1$, $\dim(h(\mathcal{M}^{[1,1+q]})) > 1$. Then there exist a unique $W \in \mathbb{R}^{p \times q}$ and $\boldsymbol{b} \in \mathbb{R}^p$ such that*

$$h(\Upsilon) = \begin{bmatrix} \Upsilon_1 \\ W\boldsymbol{\Upsilon_2} + (1 - e^{-\Upsilon_1})\boldsymbol{b} \end{bmatrix}. \quad (12)$$

*Conversely, for every pair $(W, \boldsymbol{b}) \in \mathbb{R}^{p \times q} \times \mathbb{R}^p$, the map $h$ defined by equation 12 is a homomorphism.*

The proof of the theorem is in Appendix E, and relies on defining the homomorphisms on the algebra generators. Notice that we can also use a non-square $W$ to change the manifold dimension.

**General linear layer.** We want to define the linear layer as a composition of homomorphisms from a solvable group to the next one and isometries from the group to itself, as discussed in Sec. 2. By combining Eq. 10-12, we find the hyperbolic linear layer as the transformation $f_{\text{lin}} : \mathcal{M}^{[1, 1+q]} \to \mathcal{M}^{[1, 1+r]}$ given by

$$f_{\text{lin}}(\Upsilon) = R_{\boldsymbol{u}}\left(\begin{bmatrix} \beta_1 \\ \boldsymbol{\beta_2} \end{bmatrix} * \begin{bmatrix} \Upsilon_1 \\ W\boldsymbol{\Upsilon}_2 + \boldsymbol{b} \end{bmatrix}\right), \quad (13)$$

where $W \in \mathbb{R}^{r \times q}$, $\boldsymbol{b} \in \mathbb{R}^r$, $\beta \in \mathcal{M}^{[1, 1+r]}$ and $\boldsymbol{u} \in \mathbb{S}^{r+1}$, which are the parameters that are learned during training. Notice that the orthogonal matrix $Q$ of Eq. 10 has been absorbed in the matrix $W$.

Our formulation of hyperbolic layers is different from previous iterations (Ganea et al., 2018; Shimizu et al., 2021), which rely on Riemannian logarithmic and exponential maps. The hyperbolic linear layers are usually defined as

$$y = \exp_b\left(\mathbf{P}_{\mathbf{0} \to b} W \log_{\mathbf{0}}(x)\right), \quad (14)$$

where $\exp_b : T_b\mathcal{M} \to \mathcal{M}$ is the Riemannian exponential map in the point $b \in \mathcal{M}^{[1,1+q]}$, $\log_{\mathbf{0}} : \mathcal{M} \to T_{\mathbf{0}}\mathcal{M}$ is the Riemannian logarithmic map in the origin, $\mathbf{P}_{\mathbf{0} \to b}$ is the parallel transport from $\mathbf{0}$ to $b$, and $W \in \mathbb{R}^{(q+1) \times (q+1)}$.

As any $\varphi \in \text{Iso}(\mathcal{M}^{[1,1+q]})$ can be written (from the Cartan–Ambrose–Hicks theorem (Cheeger, 1975) through the Riemannian exponential map substituting $W$ with $Q \in \text{SO}(1 + q)$ in Eq. 14, we find that existing architectures parametrize all the isometries of the space. However, since $W$ is a generic linear operation on the coordinates, it is a generic nonlinear operation on the algebra, and hence breaks the symmetries between layers.

Each application of a hyperbolic linear layer (Eq. 13) mixes the Cartan coordinate and the fiber coordinates through the fiber rotation. The first coordinate of Eq. 11 is then exponentiated in the following layer, adding nonlinearities to the expression, so stacking hyperbolic layers increases expressivity even without the addition of an activation function.

### 3.3 HYPERBOLIC SOFTMAX

**Hyperbolic hyperplanes.** In analogy to Euclidean space, we consider the set of geodesically complete submanifolds that separate $\mathcal{M}^{[1,\,1+q]}$ into two halves. These manifolds are the same subspaces as the Poincaré hyperplanes of Ganea et al. (2018); Shimizu et al. (2021) and are introduced as geodesically convex hulls in Chami et al. (2021). They are given by all possible isometric immersions of $\mathcal{M}^{[1,\,q]}$ into $\mathcal{M}^{[1,\,1+q]}$.

The general equation for these hyperplanes in solvable coordinates is as follows:

$$
\begin{aligned}
\mathrm{H}_{\alpha,\beta,\boldsymbol{w}} = \{\Upsilon \in \mathcal{M}^{[1,\,1+q]} \ s.t. \\
h_{\alpha,\beta,\boldsymbol{w}}(\Upsilon) = \alpha\, e^{-\Upsilon_1} + \langle \boldsymbol{w},\, \boldsymbol{\Upsilon}_2 \rangle + \beta\, e^{\Upsilon_1}\left(1 + \|\boldsymbol{\Upsilon}_2\|^2\right) = 0\} \\
\text{with: } \|\boldsymbol{w}\|^2 - 4\,\alpha\,\beta > 0, \quad \alpha,\,\beta \in \mathbb{R},\ \boldsymbol{w} \in \mathbb{R}^q
\end{aligned}
\tag{15}
$$

where $\langle\,,\,\rangle$ is the Euclidean scalar product. For details on the derivation, refer to Appendix F.

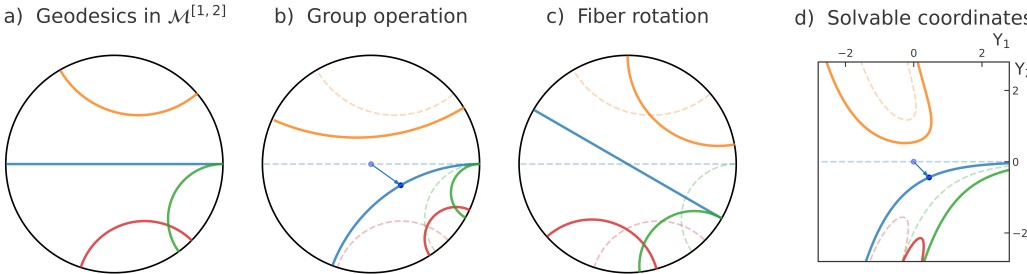

a) Geodesics in $\mathcal{M}^{[1,\,2]}$    b) Group operation    c) Fiber rotation    d) Solvable coordinates

Figure 2: **Hyperplanes in $\mathcal{M}^{[1,2]} \simeq \mathbb{H}^n$.** This figure illustrates an example of the hyperplanes that divide the hyperbolic space. For $q = 1$, they correspond to the set of all the geodesics. (a) In the Poincarè disk model, the geodesics consist of all arcs of Euclidean circles orthogonal to the disk boundary, plus all the disk diameters. (b) Geodesics obtained by applying the isometry given by left multiplication (Eq. 7) to the whole space. (c) Geodesics obtained by applying a fiber rotation (Eq. 11) (d). The same geodesics as b) in solvable coordinates.

**Logistic regression layer.** The general formula for logistic regression in hyperbolic space is

$$
p(y = 1\,|\,\Upsilon) = \hat{y}(\Upsilon) = \sigma\left(h_{\alpha,\beta,\boldsymbol{w}}(\Upsilon)\right).
\tag{16}
$$

The distance of a point $\Upsilon$ from a generic separator is

$$
d(\Upsilon, H_{\alpha,\beta,\boldsymbol{w}}(\Upsilon)) = \frac{1}{2}\mathrm{arccosh}\left(1 + 2\,\frac{h^2_{\alpha,\beta,\boldsymbol{w}}(\Upsilon)}{\|\boldsymbol{w}\|^2 - 4\alpha\beta}\right),
\tag{17}
$$

where the distance from a subspace $\mathcal{S}$ is defined as $d(\Upsilon, \mathcal{S}) = \min_{\Psi \in \mathcal{S}} d(\Upsilon, \Psi)$. The argument of the sigmoid in eq. 16 is then a nonlinear monotonic function of the distance between each point and the hyperplane (notice the subtle difference from the Euclidean case). Similarly, when classifying between $K$ classes in hyperbolic space, we can define the analogous hyperbolic softmax regression as

$$
p(y = k\,|\,\Upsilon) = \frac{\exp\left(h_k(\Upsilon)\right)}{\sum_{j=1}^{K} \exp\left(h_j(\Upsilon)\right)},
\tag{18}
$$

where $h_j(\Upsilon) = h_{\alpha_j,\beta_j,\boldsymbol{w}_j}(\Upsilon)$.

### 3.4 HYPERBOLIC CARTAN NETWORKS

We construct the simplest hyperbolic neural network by stacking $L$ hyperbolic linear layers such that

$$h^\ell = R_{\boldsymbol{u}^\ell} \left( \beta^\ell * \begin{bmatrix} h_1^{\ell-1} \\ W^\ell \boldsymbol{h}_2^{\ell-1} + \boldsymbol{b}^\ell \end{bmatrix} \right), \tag{19}$$

and predicting on the $L$-th layer representations through the logistic layer (binary classification) or the logistic softmax (multiclass classification).

We also propose that the initial embedding of the starting data points $\boldsymbol{x_i} \in \mathbb{R}^d$ into the first hyperbolic layer $\mathcal{M}^{[1,\,1+d]}$ is as follows:

$$h^1 = \begin{bmatrix} 0 \\ \boldsymbol{x} \end{bmatrix}. \tag{20}$$

Notice that by setting $\boldsymbol{u}'^\ell = \boldsymbol{0}$, $\beta_1^\ell = 0 \, \forall \ell$ this architecture becomes a stack of Euclidean linear layers.

**Universal approximation properties.** A composition of Cartan layers is not a universal approximator: its expressivity is at most polynomial in the input variables, with order depending on network depth. In contrast to Euclidean linear layers, however, stacking these hyperbolic layers creates an increasingly more expressive function class.

Given a pointwise nonlinearity $\sigma : \mathbb{R} \to \mathbb{R}$, we can apply it to our coordinates by

$$\sigma(\Upsilon) = \begin{bmatrix} \Upsilon_1 \\ \sigma(\boldsymbol{\Upsilon}_2) \end{bmatrix}. \tag{21}$$

Cartan networks with such nonlinearities are universal approximators. In fact, from Eq. 19, the choice of $\beta_1 = 0$, $\boldsymbol{u} = (1, 0, \dots, 0)$ removes all nonlinearities deriving from the hyperbolic nature of the layers. For this particular choice of parameters, the hyperbolic linear layer reverts to a fully connected Euclidean linear layer in the fiber coordinates, so the functional class of Cartan networks includes that of Euclidean neural networks, and hence inherits all the universal approximation results applicable to linear layers with activation functions. This application of nonlinearities is conceptually different from iterations of hyperbolic networks that applied nonlinearities to the tangent spaces (Peng et al., 2022; Fan et al., 2022).

**Architectural flexibility.** Cartan networks preserve the architectural flexibility of other hyperbolic architectures, as it is possible to impose the convolutional bias at the homomorphism level, thus achieving a function class with translation invariance that still extends the Euclidean convolutional neural network, as detailed in Appendix G. Using this grading of coordinates, it is possible to implement layers incorporating other architectural biases, such as batch normalization, dropout, and pooling. Much like activation functions, a naive but effective approach we take in this paper is to perform these operations by applying them only to the fiber coordinates. Better versions of these operations, accounting for the geometry of the space, could be developed by reiterating their design from their functional principles, in the solvable manifold (e.g., the running mean of batch normalization realized with geodesic averages).

These architectures can then be optimized on traditional loss functions (such as MSE and categorical cross-entropy) using Riemannian SGD or Adam (Bonnabel, 2013; Becigneul & Ganea, 2019). We will discuss optimization in depth in Appendix H.

## 4 RESULTS

We compare the performance of hyperbolic Cartan networks trained on real datasets with that of traditional neural networks and other hyperbolic neural networks (the datasets and models are discussed in Appendices I.1 and I.2). Notice that the comparison is warranted given that an Euclidean

Table 1: Accuracy on real-world datasets (mean $\pm$ std, $n = 5$)

| Problem | Cartan | Euclidean | Hyperbolic++ | Fully Hyperbolic | Poincaré |
|---------|--------|-----------|--------------|------------------|----------|
| Cifar10 | $52.6 \pm 0.3$ | $52.6 \pm 0.5$ | $52 \pm 1$ | $52.4 \pm 0.8$ | $52.5 \pm 0.3$ |
| FMNIST | $89.3 \pm 0.3$ | $89.3 \pm 0.1$ | $87.9 \pm 0.5$ | $89.2 \pm 0.2$ | $89.4 \pm 0.2$ |
| KMNIST | $90.10 \pm 0.07$ | $90.0 \pm 0.1$ | $89 \pm 1$ | $90.29 \pm 0.10$ | $90.2 \pm 0.2$ |
| MNIST | $98.27 \pm 0.02$ | $98.27 \pm 0.02$ | $98.0 \pm 0.1$ | $98.14 \pm 0.04$ | $98.19 \pm 0.06$ |

fully-connected layer ($Wx+b$) from $n$ to $m$ neurons has $m(n+1)$ parameters, while a Lie hyperbolic linear layer $\mathcal{M}^{[1,n]} \to \mathcal{M}^{[1,m]}$ has $m(n+1) - 1$ parameters. Given this, we compare networks with the same number of layers and the same size. A brief comparison of the number of operations in Cartan layers versus Euclidean ones is provided in Appendix I.3.

To characterize the performance of the proposed architecture, we train fully-connected Cartan networks on the real-world classification datasets, varying depth (1-4 layers) and hidden layer size (20, 40, 100, and 200 neurons), with and without nonlinearities, comparing their test accuracy to Euclidean and different hyperbolic networks. The best accuracies across configurations for these tasks are shown in Tab. 1, while training parameters and computation time are discussed in Appendices I.4 and I.5. We underline the model with the highest performance, and any others whose results are statistically equivalent to it. If no model is underlined, it means that no statistically significant best performer could be identified. Our proposed architecture achieves performance comparable to the alternatives across a wide range of hyperparameters, matching or surpassing the Euclidean variant.

To demonstrate the capacity of our framework to generalize to convolutional architectures, we test our architecture on a hyperbolic variant of AlexNet (Krizhevsky et al., 2012), where we replace all layers with their Cartan counterparts. This construction mirrors the original AlexNet design in terms of depth, filter sizes, and overall structure, ensuring that any performance differences can be attributed to the shift from Euclidean to hyperbolic representations. We discuss the technical details in more depth in Appendix G. We then benchmark this Cartan AlexNet against the standard Euclidean AlexNet on some real-world datasets, and we summarize the results in Tab. 2. The results show that adding hyperbolic flexibility to established architectures can improve their performance on moderately complex tasks. We additionally test ResNet and its hyperbolic variant, where the group operation takes the place of the residual connection, on the TinyImagenet dataset.

Table 2: Test accuracy (%) for AlexNet and ResNet (mean $\pm$ std, $n_{\text{runs}} = 5$)

| Problem | Alexnet | H-Alexnet |
|---------|---------|-----------|
| CelebA | $77.8 \pm 0.5$ | $77.4 \pm 0.7$ |
| Cifar10 | $88.42 \pm 0.09$ | $88.4 \pm 0.5$ |
| Cifar100 | $54.4 \pm 0.3$ | $59.5 \pm 0.8$ |
| TinyImagenet | $38.1 \pm 0.7$ | $44.6 \pm 0.3$ |

| Problem | ResNet18 | H-ResNet18 |
|---------|----------|------------|
| TinyImagenet | $61.4 \pm 0.2$ | $61.5 \pm 0.1$ |

## 5 DISCUSSION

This work introduced *Cartan networks*, a novel hyperbolic deep learning architecture based entirely on intrinsic group-theoretical and Riemannian operations. The long-term goal of this new theoretical construction is to develop a framework for machine learning algorithms that is both expressive and mathematically consistent, resulting in models that can be more easily analyzed and interpreted.

Cartan networks complement the view of hyperbolic neural networks as a sequence of exponential and logarithmic maps (Ganea et al., 2018). Our architecture exploits the dual nature of hyperbolic spaces as solvable groups and Riemannian manifolds, alternating isometries and homomorphisms in its layers, both intrinsically defined and geometrically motivated operations.

The experiments performed, although not exhaustive, demonstrate that the proposed architecture is competitive with comparable Euclidean and hyperbolic architectures on a range of real and synthetic tasks. The performance of hyperbolic convolutional architectures demonstrates the approach's flexibility and scalability. These results are particularly encouraging because the hyperbolic space is the simplest representative of the family of non-compact symmetric spaces; exploring more complex manifolds and investigating how to specialize existing layers to hyperbolic geometry are major directions for future research.

While our results demonstrate the potential of the proposed framework, several limitations must be acknowledged. First, due to resource constraints, our experiments were conducted on a limited number of datasets and with a relatively narrow range of hyperparameter configurations. Secondly, the architectural modifications introduced to incorporate group-theoretical structure lead to increased computational overhead. While this is somewhat inevitable given the high optimization of standard neural network software, improving the computational performance of our approach is an important step to ensure its adoption.

## 6 REPRODUCIBILITY STATEMENT

The code used for the experiments in this work is based on PyTorch, an open-source deep learning Python library (Paszke et al., 2019). Optimization routines, particularly those involving geometry-aware methods, utilize the Geoopt library (Kochurov et al., 2020). The entire code to reproduce all the results shown in this article is available at

https://github.com/CartanNetworks/CartanNetworks

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

## APPENDIX A  LIE GROUPS

A Lie group (Helgason, 1962; Fré, 2023; Magnea, 2002; Humphreys, 1972) is an analytic differentiable manifold $G$ endowed with a group structure such that the group operations of multiplication and inversion are infinitely differentiable. The group operation is a binary product operation:

$$* : G \times G \to G \quad (x * y) = xy \in G, \tag{22}$$

that must satisfy the group axioms. An abstract Lie group always admits an infinite series of matrix representations, which are determined by the abstract group structure, where the group operations are matrix multiplication and inversion. In practice, Lie groups are both groups (having multiplication and inverses) and smooth manifolds (having a differentiable structure). Every Lie group has a corresponding Lie Algebra isomorphic as a vector space to its tangent space at the identity. It is formed by its (left/right) invariant vector fields that close under commutation. On Lie groups $G$ and on coset manifolds $G/H$, one can construct $G$-invariant Riemannian metrics that are unique or multiple depending on the structure of the coset.

## APPENDIX B  SOLVABLE COORDINATES PARAMETRIZATION

As introduced in Bruzzo et al. (2025), the matrix element parameterizing the hyperbolic space is as follows[1]:

$$\mathbb{L}(\Upsilon) = \begin{bmatrix} e^{\Upsilon_1} & \sqrt{2}e^{\Upsilon_1}\boldsymbol{\Upsilon_2}^{\mathsf{T}} & -e^{\Upsilon_1}|\boldsymbol{\Upsilon_2}|^2 \\ 0 & \mathbb{I}_q & -\sqrt{2}\boldsymbol{\Upsilon_2} \\ 0 & 0 & e^{-\Upsilon_1} \end{bmatrix}, \tag{23}$$

where

$$\Upsilon = [\Upsilon_1, \boldsymbol{\Upsilon_2}]^{\mathsf{T}} = [\Upsilon_1, \Upsilon_{2,1}, \ldots, \Upsilon_{2,q}]^{\mathsf{T}} \tag{24}$$

---

[1]Compared to the original formulation of Fré et al., we performed the following change of coordinates $\boldsymbol{\Upsilon}_2 = \frac{1}{2}\boldsymbol{\Upsilon}_2^{\text{old}}$ for ease of formulation.

are the solvable coordinates, $\mathbb{I}_q$ is the identity matrix of size $q$, and $|.|$ is the Euclidean norm. Group operation is given by matrix multiplication:

$$\mathbb{L}(\Psi * \Upsilon) = \mathbb{L}(\Psi) \cdot \mathbb{L}(\Upsilon). \tag{25}$$

In the upper-triangular representation, the matrices preserve the Lorentz metric

$$\eta_j^i = \delta_{i,n-j}, \quad i,j = 1,\ldots n. \tag{26}$$

In these coordinates, by left-transport of the metric induced on the solvable Lie algebra at the origin by the Einstein metric of the symmetric space, we find

$$g_{1,q}(\Upsilon) = \begin{bmatrix} 1 + |\mathbf{\Upsilon_2}|^2 & \mathbf{\Upsilon_2^\intercal} \\ \mathbf{\Upsilon_2} & \mathbb{I}_q \end{bmatrix}. \tag{27}$$

Notice that in these coordinates, the volume element is constant:

$$\sqrt{\det g_{1,q}(\Upsilon)} = 1. \tag{28}$$

**Transition to the Poincaré Ball coordinates.** In this section, we provide the transition function from the solvable coordinates to the Poincaré Ball coordinates, namely the projective off-diagonal coordinates of Gilmore (2016) and Equation (5.2.43) of Fré (2012). Given a point in $\mathbb{H}^{q+1}$ labeled Poincaré ball coordinates $x$, the same point is identified by a set of solvable coordinates $\Upsilon$ related to $x$ in the following way. First, we split the coordinates as follows:

$$x = [x_1, \mathbf{x_2}]^\intercal = [x_1, x_{2,1}, \ldots, x_{2,q}]^\intercal. \tag{29}$$

Then, the map from the solvable parametrization to the point in the Poincaré ball is given by

$$\begin{cases} x_1 = 1 - \dfrac{1 + e^{-\Upsilon_1}}{1 + \cosh N(\Upsilon)} \\ \mathbf{x_2} = \dfrac{\mathbf{\Upsilon_2}}{1 + \cosh N(\Upsilon)} \end{cases} \tag{30}$$

## APPENDIX C   RIEMANNIAN OPERATIONS IN SOLVABLE COORDINATES

**Distance between points.** Given $\Upsilon \in \mathcal{M}^{[1,1+q]}$, its norm is

$$N(\Upsilon) = \operatorname{arccosh}\left(\frac{1}{2}(e^{-\Upsilon_1} + e^{\Upsilon_1}(1 + \|\mathbf{\Upsilon_2}\|^2))\right). \tag{31}$$

Then, for $\Upsilon, \Psi \in \mathcal{M}^{[1,q]}$, their distance is given by $d(\Upsilon, \Psi) = N(\Psi^{-1} * \Upsilon)$.

**Vector transport.** The group operation on $\mathcal{M}^{[1,1+q]}$ naturally induces a notion of transport. Specifically, the left group action $L_{\Psi * \Upsilon^{-1}}$ defines a diffeomorphism $dL_{\Psi \Upsilon^{-1}} : T_\Upsilon \mathcal{M} \to T_\Psi \mathcal{M}$ that acts on a tangent vector $v \in T_\Upsilon \mathcal{M}$ as

$$dL_{\Psi * \Upsilon^{-1}}(v) = \begin{bmatrix} v_1 \\ \mathbf{v_2} - v_1(\mathbf{\Psi_2} - \mathbf{\Upsilon_2}) \end{bmatrix}. \tag{32}$$

**Riemannian logarithmic map.** In general, the Riemannian logarithmic map can be retrieved from the geodesic distance equation from the following formula (do Carmo, 1992):

$$\nabla_{R,\mathbf{\Psi}}(d^2(\mathbf{\Upsilon}, \mathbf{\Psi})) = -2\log_{\mathbf{\Psi}} \mathbf{\Upsilon}, \tag{33}$$

where $\nabla_R f = g^{-1} df$ denotes the Riemannian gradient.

In solvable coordinates, then,

$$\log_0(\Upsilon) = \frac{N(\Upsilon)}{\sinh N(\Upsilon)} \left[ \begin{matrix} \cosh N(\Upsilon) - e^{-\Upsilon_1} \\ \Upsilon_2 \end{matrix} \right]. \tag{34}$$

At a general point $\Psi \in \mathcal{M}$, we can also use the left-invariance to compute the logarithmic map. That is, we first translate to $\Psi$ the origin, apply $\log_0$, and then use the inverse parallel transport to bring the result back to the tangent space at $\Psi$:

$$\log_\Psi(\Upsilon) = dL_{\Psi*0}^{-1} \left( \log_0(\Psi^{-1} * \Upsilon) \right). \tag{35}$$

**Geodesics.** The geodesic can be computed with the general method described in Bruzzo et al. (2025). Given a tangent vector $\boldsymbol{v} = \{v_1, \boldsymbol{v}_2\} \in T_0\mathcal{M}$ and its norm $\|\boldsymbol{v}\| := \sqrt{\sum_i v_i^2}$, the formula for the geodesics from the origin is

$$\boldsymbol{\gamma}_0(\boldsymbol{v}, t) = \left[ \begin{matrix} -\log\left( \cosh(\|\boldsymbol{v}\| t) - \dfrac{v_1}{\|\boldsymbol{v}\|} \sinh(\|\boldsymbol{v}\| t) \right) \\ \dfrac{\boldsymbol{v}_2}{\|\boldsymbol{v}\|} \sinh(\|\boldsymbol{v}\| t) \end{matrix} \right], \tag{36}$$

with $t \in [0, 1]$.

Then the geodesic between points $\Upsilon, \Psi \in \mathcal{M}^{[1, 1+q]}$ is obtained by applying the logarithmic map $\log_0$ to $\Upsilon * \Psi^{-1}$, tracing the geodesic $\boldsymbol{\gamma}_0$, and translating back using the group action:

$$\boldsymbol{\gamma}_{\Psi \to \Upsilon}(t) = \Psi * \boldsymbol{\gamma}_0 \left( t, \log_0(\Psi^{-1} * \Upsilon) \right). \tag{37}$$

**Exponential Riemannian map.** From Eq. 36, we find that the exponential map from the origin is

$$\exp_0(\boldsymbol{v}) = \boldsymbol{\gamma}_0(\boldsymbol{v}, t=1) = \left[ \begin{matrix} -\log\left( \cosh(\|\boldsymbol{v}\|) - \dfrac{v_1}{\|\boldsymbol{v}\|} \sinh(\|\boldsymbol{v}\|) \right) \\ \dfrac{\boldsymbol{v}_2}{\|\boldsymbol{v}\|} \sinh(\|\boldsymbol{v}\|) \end{matrix} \right], \tag{38}$$

for $\boldsymbol{v} \in T_0\mathcal{M}$. The generic exponential map is then

$$\exp_\Upsilon(\boldsymbol{v}) = \Upsilon * \exp_0(dL_{0\Upsilon^{-1}}(\boldsymbol{v})). \tag{39}$$

# APPENDIX D   ISOMETRIES

## D.1   RELEVANT ISOMETRIES FROM GROUP THEORY

The coset manifold $\mathcal{M}^{[1,q+1]}$ is defined as a quotient $\dfrac{\mathrm{SO}(1, q+1)}{\mathrm{SO}(q+1)}$ and metrically equivalent to a group $\mathrm{Exp}(\mathrm{Solv}_{1,q+1})$. Its isometries are all the transformations of the group $\mathrm{SO}(1, q+1)$, which we will classify into two groups.

1. The multiplication by a solvable group element.
2. The adjoint action of the full group on the solvable group.

As per Bruzzo et al. (2025), we can split the algebra $\mathbb{H}^{[1,q+1]}$ in two different components:

$$\mathbb{H}^{[1,q+1]} = \mathbb{G}_{\mathrm{paint}}^{[1,q+1]} \oplus \mathbb{H}_F^{[1,q+1]}. \tag{40}$$

We refer to the exponential of the first component as the paint group, while the second component corresponds to the fiber rotation. Together with the solvable element multiplication, these form the three categories in which we split the full algebra.

## D.2 EXPLICIT DERIVATION OF ISOMETRIES IN THE PGTS COORDINATES

The set of isometries (distance-preserving maps) of $\mathcal{M}^{[1\,1+q]}$ into itself is given by $\mathrm{SO}(1, 1 + q)$ (these have been parameterized in terms of the Poincarè ball coordinates by Jaćimović (2025)). These isometries are a composition of three distinct isometries (for a detailed derivation, refer to Bruzzo et al. (2025)).

**Paint rotation.** The group of outer automorphisms (within the full isometry group $\mathrm{SO}(1, 1 + \mathrm{q})$) of the solvable Lie group $\mathcal{S}$ metrically equivalent to our symmetric space corresponds to the notion of *Paint Group* originally introduced in Fré et al. (2006) and fully discussed in Bruzzo et al. (2025). It is named $\mathcal{G}_{\mathrm{paint}}$. For $r = 1$, $\mathcal{G}_{\mathrm{paint}} \sim \mathrm{SO}(q)$, and each $Q \in \mathrm{SO}(q)$ maps a point with solvable coordinates $\Upsilon$ by rotating $\boldsymbol{\Upsilon}_2$:

$$\begin{cases} \Upsilon_1^{\mathrm{paint}} = \Upsilon_1 \\ \boldsymbol{\Upsilon_2^{\mathrm{paint}}} = Q\boldsymbol{\Upsilon_2} \end{cases} \tag{41}$$

**Group translation.** Each element $b \in \mathcal{M}^{[1, 1+q]}$ defines an isometry of the symmetric space into itself through the group action. From the geometric point of view, this represents a rigid translation of the origin $\mathbf{0}$ into point $b$. This operation will take the role of the *bias* of classical logistic regression.

**Fiber rotation.** The full group of outer automorphisms of $\mathrm{G/H}$ is given by the exponential of $\mathbb{H} = \mathbb{G}_{\mathrm{paint}} \oplus \mathbb{H}_{\mathrm{F}}$. (see Bruzzo et al. (2025) for the theory of the non-compact symmetric space Grassmannian foliation to which the Lie subalgebra $\mathbb{G}_{\mathrm{F}} \subset \mathfrak{so}(1, 1 + q)$ is tightly connected). By means of the exponential map the subalgebra $\mathbb{G}_{\mathrm{F}} \subset \mathfrak{so}(1, 1 + q)$ generates a $q$-dimensional group of isometries. Each of these isometries modifies the Cartan coordinate $\Upsilon_1$ and coordinate $\Upsilon_{2,j}$.

To derive an analytic expression, however, we use the fact that isometries of Riemannian manifolds can be parametrized in terms of the exponential map. In particular, as paint rotations are given by matrices $Q \in \mathrm{SO}(q)$, the remaining isometries are parametrized by the generators of the full group $\mathrm{SO}(q + 1)$ without the paint generators $\mathrm{SO}(q)$, and can be computed accordingly.

Given a vector $\boldsymbol{u} = [u_0, u_1, \ldots, u_q]^\mathsf{T} \in \mathbb{S}^{q+1}$ ($|\boldsymbol{u}| = 1$), and defining $\boldsymbol{u}' = [u_1, \ldots, u_q]^\mathsf{T}$, the total fiber rotation by $\boldsymbol{u}$ is given by

$$R_u(\Upsilon) = \begin{bmatrix} -\log\left(-\dfrac{1}{2}(\mathrm{e}^{\Upsilon_1}(1 + \|\boldsymbol{\Upsilon}_2\|^2) + \mathrm{e}^{-\Upsilon_1})(1 + u_0) + \mathrm{e}^{-\Upsilon_1}u_0 - \boldsymbol{\Upsilon}_2 \cdot \boldsymbol{u}'\right) \\ \boldsymbol{\Upsilon}_2 - x\left(\dfrac{\boldsymbol{\Upsilon}_2 \cdot \boldsymbol{u}'}{1 + u_0} + \dfrac{1}{2}(\mathrm{e}^{\Upsilon_1}(1 + \|\boldsymbol{\Upsilon}_2\|^2) - \mathrm{e}^{-\Upsilon_1})\right)\boldsymbol{u}' \end{bmatrix}. \tag{42}$$

A general isometry $f : \mathcal{M}^{[1, 1+q]} \to \mathcal{M}^{[1, 1+q]}$ can be parametrized as

$$f(\Upsilon) = R_{\boldsymbol{u}}\left(\begin{bmatrix} b_1 \\ \boldsymbol{b}_2 \end{bmatrix} * \begin{bmatrix} 1 & 0 \\ 0 & Q \end{bmatrix} \begin{bmatrix} \Upsilon_1 \\ \boldsymbol{\Upsilon}_2 \end{bmatrix}\right), \tag{43}$$

where $Q \in \mathrm{SO}(q)$, $b \in \mathcal{M}^{[1, 1+q]}$ and $\boldsymbol{u} \in \mathbb{S}^{q+1}$.

## APPENDIX E   HOMOMORPHISMS

In this section, we prove Th. 3.1.

*Proof.* Let $h$ be an homomorphism between $\mathcal{M}^{[1,q+1]}$ and $\mathcal{M}^{[1,p+1]}$. Since they are both simply connected, Th. 5.6 from Hall (2015) applies, hence there exists a unique Lie Algebra morphism $\mathfrak{h} : \mathrm{Lie}(\mathcal{M}^{[1,q+1]}) \to \mathrm{Lie}(\mathcal{M}^{[1,p]+1})$ such that $\mathfrak{h} = \mathrm{d}h$. To find all such morphisms, it is enough to parametrize all algebra homomorphisms $\mathfrak{h}$.

Since these homomorphisms are vector space morphisms, it is enough to define them on algebra generators. The generators are given in Bruzzo et al. (2025) and satisfy the following relationships:

$$[H, T_i] = T_i, \quad [T_i, T_j] = 0.$$

Let $H^q, T_i^q$ be the generators of $\mathcal{M}^{[1,q]}$ and $H^p, T_i^p$ be the generators of $\mathcal{M}^{[1, 1+p]}$. It is enough to find linear maps that satisfy the commutator relations, that is

$$[\phi(H^q), \phi(T_i^q)] = \phi(T_i^q),$$

as all other relations will not give additional constraints. By setting

$$\phi(H^q) := \alpha H^p + \beta^i T_i^p, \quad \phi(T_j^q) := \alpha_j H^p + W_j^i T_i^p,$$

one can check the commutators for all generators, thus obtaining

$$[\phi(H^q), \phi(T_j^q)] = [\alpha H^p + \beta^i T_i^p, \alpha_j H^p + W_j^l T_l]^p = \alpha W_j^l T_l^p - \beta^i \alpha_j T_i^p = \phi(T_j^q) = \alpha_j H^p + W_j^m T_m^p.$$

from which $\alpha_j = 0$. As the dimension of the image is greater than 1 by assumption, at least one $T_j^q$ must have a nontrivial image, hence $\alpha = 1$. Hence, the homomorphism matrix in the basis of these generators is given by

$$\tilde{W} = \begin{bmatrix} 1 & 0 \\ \boldsymbol{\beta} & W \end{bmatrix}.$$

All that remains is to express these morphisms in terms of solvable coordinates. The relationship between solvable coordinates and algebra coordinates is given by the map $\chi$:

$$\chi\left(\begin{bmatrix} t^1 \\ \boldsymbol{t^2} \end{bmatrix}\right) = \begin{bmatrix} \Upsilon_1 \\ \dfrac{\Upsilon_1}{1 - e^{-\Upsilon_1}} \boldsymbol{\Upsilon_2} \end{bmatrix}.$$

Then, our group element with coordinates $\Upsilon = \chi(t)$ is written as

$$\mathbb{L}(\chi(t)) = \text{Exp}(t^1 H + t^i T_i),$$

and the homomorphism in coordinates is the map

$$\mathfrak{h} = \chi \circ \tilde{W} \circ \chi^{-1},$$

which after trivial manipulation gives Eq. 12. $\square$

*Remark.* Although the abstract exponential map from a Lie algebra to the component connected to the Identity of a corresponding Lie group is unique, its explicit realization in terms of *group parameters* namely, coordinates on the group manifold depend on the definition of the atlas of open charts and can then take many different forms. Since the solvable group S and hence its metric equivalent non-compact symmetric space U/H are diffeomorphic to $\mathbb{R}^n$, we have just one open chart that covers the entire non-compact manifold. However, this open chart, namely the utilized solvable coordinates, can be chosen in several different ways, depending on the way the exponential map $\Sigma : Solv \to \mathcal{S}$ is done matrix-wise. As explained in Bruzzo et al. (2025), for the *normed solvable Lie Algebras* uniquely associated to each n.c. G/H, the generators that are in one-to-one relations with the TS projection of the G root system have a natural grading in terms of root heights, and this introduces a canonical definition of the $\Sigma$ exponential map that is the one adopted in the present paper. The relation between the canonical solvable coordinates $\boldsymbol{\Upsilon}_i$ of the $i$-th solvable group $\mathcal{S}_i$ and those $\boldsymbol{\Upsilon}_{i+1}$ of its homomorphic image $\mathcal{S}_{i+1}$ generated by the linear homomorphism of the corresponding solvable Lie algebras can be obtained by solving the first-order differential system provided by the linear relation between Maurer Cartan 1-forms. Such a system is always iteratively solvable by quadratures precisely because the Lie algebras are solvable.

## Appendix F  Derivation of hyperbolic hyperplanes

In the hyperbolic space $\mathcal{M}^{[1, 1+q]}$, the set of submanifolds of codimension 1 is given by all possible immersions of $\mathcal{M}^{[1, q]}$ (Kobayashi & Nomizu, 1963).

These hyperplanes can be found by defining one such immersion, for example

$$\mathbf{H_0}^{1+q} = \{\Upsilon \in \mathcal{M}^{[1,\,1+q]} \mid \Upsilon_{2,q} = 0\} \simeq \mathcal{M}^{[1,\,q]}, \tag{44}$$

and finding the set of isometries that do not leave $\mathbf{H_0}$ invariant. Given the complete isometry group $G_{q+1}$ of the manifold $\mathcal{M}^{[1,q+1]}$, embedding $\mathcal{M}^{[1,q]} \hookrightarrow \mathcal{M}^{[1,q+1]}$ also gives an injective homomorphism $G_q \hookrightarrow G_{q+1}$, so the set of isometries we look for is the quotient $G_{q+1}/G_q$. The isometry categories, given in Appendix D.2, all have easily recognizable realizations in the quotient. For the paint rotation, we consider the rotations of the $q$-th paint coordinate onto the others, that is, the $q$-sphere $SO(q)/SO(q-1)$. For the fiber rotation, we consider the one-parameter subgroup generated by rotating the $q$-th coordinate. Since the points $\Psi \notin \mathbf{H_0}^{1+q}$ map the fundamental separator into a different separator, the remaining isometries can be thought without loss of generality as the group action of the points $\Psi \in \mathcal{M}^{[1,\,1+q]}$ with solvable coordinates

$$\Psi = [0,\, 0,\, \ldots, 0,\, \Psi_{2,q}]^{\mathsf{T}}. \tag{45}$$

We obtain Eq. 15 by combining these three isometries. In $\mathcal{M}^{[1,\,1+q]}$, totally geodesic hyperplanes can also be characterized as sets of points $\{\Upsilon \in \mathcal{M}^{[1,\,1+q]} \; s.t. \; \langle w, \log_\Psi(\Upsilon)\rangle = 0\}$, where $\log_\Psi$ is the logarithmic map at a fixed base point $\Psi \in \mathcal{M}^{[1,\,1+q]}$, and $w \in T_\Psi \mathcal{M}^{[1,\,1+q]}$ is a fixed vector. Indeed, we can also obtain Eq. 15 from Eq. 35 and this definition of hyperplanes.

The distance between a point and the submanifold $\mathbf{H}_0^{1+q}$ only depends on its $q$-th coordinate and is easily obtained by minimization and given by

$$d(\Upsilon, \pi_0) = \frac{1}{2}\text{arccosh}\left(1 + 2\,\Upsilon_{q+1}^2\right). \tag{46}$$

Since every regression separator is the image of the subspace $\mathbf{H}_0^{1+q}$ through an isometry $\Phi$, $h_{\alpha,\beta,\boldsymbol{w}}(\Upsilon)$ in Eq. 15 is proportional to the $q$-th coordinate of $\Phi(\Upsilon)$. The proportionality factor is $(\|\boldsymbol{w}\|^2 - 4\alpha\beta)^{-1}$, and from this we obtain Eq. 16.

## APPENDIX G   CONVOLUTIONAL ARCHITECTURES

To implement a hyperbolic version of the convolution operation, we make the following considerations:

- We treat each image as a point in a $N_{\text{channels}} \times N_{\text{pixels}} + 1$ dimensional hyperbolic Space.
- The linear convolutional operation replaces the linear operation of the hyperbolic layer.
- Following traditional CNN implementation, bias and rotation parameters related to each channel are forced to be equal during training.
- Much like the fully connected version, the convolutional version reverts to the Euclidean variant for a trivial choice of bias and rotation parameter.

The hyperbolic version of other layers (dropout, maxpooling, local norm response) was implemented by restricting the layer action to the fiber coordinates, similarly to Eq. 21. To test the performance of our proposal on more complex tasks, we compared it against the Euclidean AlexNet architecture Krizhevsky et al. (2012). Our architecture mimicked the overall structure of AlexNet, replacing each layer with its hyperbolic counterpart. As remarked in Sec. 3.4, this is a naive way to implement these more complex architectural components in our framework, and could be later expanded to better take into account the specific geometry of the hyperbolic space.

## APPENDIX H   OPTIMIZATION

In contrast to Euclidean optimization, where gradients are computed in a flat vector space, Riemannian optimization takes into account the geometry of the manifold. Riemannian gradient methods compute gradients in this tangent space and use the retraction of the exponential maps to update parameters

back onto the manifold. In Riemannian Stochastic Gradient Descent (RSGD) (Bonnabel, 2013; Becigneul & Ganea, 2019), at each iteration $t$, the update is

$$\theta_{t+1} = \mathcal{R}_{\theta_t}(-\eta_t \nabla_R L(\theta_t)), \tag{47}$$

where $\nabla_R L(\theta_t) = g^{-1}(\theta_t) \, dL(\theta_t)$ is the Riemannian gradient, $\eta_t$ is the learning rate, and $\mathcal{R}$ is a retraction that maps the tangent space back to the manifold. Since the exact exponential map is computationally expensive, we use a first-order approximation:

$$\mathcal{R}_\theta(v) = \theta + v, \tag{48}$$

where $v \in T_\theta \mathcal{M}$. In our implementation, the Riemannian versions of SGD and Adam were provided by Geoopt (Kochurov et al., 2020). Occasionally, certain initializations lead to particularly poor training behavior, causing the loss to diverge within the first few batches. When reporting results over $N_{\text{runs}}$, we typically exclude these divergent runs, and we still do not really understand the causes of this behavior.

## APPENDIX I  NUMERICAL EXPERIMENTS

### I.1  DATASETS

**Real-world datasets.** We utilize four real-world benchmark datasets in our experiments; for these datasets, we use the standard train/test split provided by the torchvision library (Paszke et al., 2019).

- **MNIST** (LeCun et al., 1998), consisting of 70,000 grayscale images of handwritten digits (0-9) at 28x28 resolution.

- **Fashion MNIST** (Xiao et al., 2017), which contains 70,000 grayscale images (28x28 pixels) of Zalando clothing items such as shirts, trousers, and shoes.

- **K-MNIST** (Clanuwat et al., 2018), a dataset of 70,000 grayscale images (28x28 pixels) of Japanese characters from the Kuzushiji script.

- **CIFAR-10** (Krizhevsky & Hinton, 2009), composed of 60,000 color images (32x32 pixels) across ten categories, including animals (e.g., dogs, cats) and vehicles (e.g., cars, trucks).

- **CIFAR-100** (Krizhevsky & Hinton, 2009), composed of 60,000 color images (32x32 pixels) across 100 fine-grained classes grouped. The dataset provides 500 training and 100 test images per fine class.

- **CelebA** (Liu et al., 2015), a large-scale face dataset with 202,599 color images (aligned/cropped to 178×218 pixels) of 10,177 identities. Each image is annotated with 40 binary facial attributes (e.g., *Smiling*, *Wearing Glasses*) and 5 landmark locations. We built 8 categorical variables with the combination of the *Male*, *Young*, and *Smiling* attributes.

- **Tiny Imagenet** (Le & Yang, 2015), a scaled-down version of ImageNet that contains 100000 images of 200 classes (500 for each class) downsized to 64×64 colored images. Each class has 500 training images, 50 validation images, and 50 test images.

### I.2  OTHER HYPERBOLIC NEURAL NETWORKS

In this section, we describe the different hyperbolic neural networks we compared in Sec. 4, and we refer to the original articles for a detailed description of these architectures. Notice that some of these architectures have a strong focus on transformers, which are not considered in our work, so we adapted their implementation to our tasks. All the comparisons should be taken as exploratory, as an in-depth review of the performance of existing hyperbolic architectures was beyond the scope of our work.

**Poincaré coordinates.** We implemented hyperbolic neural networks based on Ganea et al. (2018) using the manifold parametrization provided by *Geoopt* (Kochurov et al., 2020) and the *HypTorch* python package (Roberts, 2025). A hyperbolic layer is given by

$$\text{Poi}_{W,b}(x) := \exp_b(P_{0 \to b} W \log_0(x)), \tag{49}$$

where $b$ is a point on the Poincaré ball, $W$ is the weight vector, and $\exp$ and $\log$ are the Riemannian exponential and logarithmic maps.

A neural network is obtained by alternating these layers and nonlinearities, with an initial embedding layer and a hyperbolic multinomial logistic regression (MLR) in the Poincaré ball as the final layer.

**Fully Hyperbolic (Lorentz).** We implemented fully hyperbolic neural networks in the Lorentz model following Chen et al. (2022), and using the code provided by Bdeir et al. (2024). Fully hyperbolic neural networks use the Lorentz model and adapt the Lorentz transformations to implement network layers. Neural networks are constructed by stacking these layers with Lorentz-compatible nonlinearities, preceded by a Lorentz embedding layer, namely the projection on the Lorentz manifold, and terminated with a Lorentz MLR.

**Hyperbolic++.** Hyperbolic networks++ extend hyperbolic architectures of Ganea et al. (2018) by reformulating the MLR head and redefining the FC layers (Shimizu et al., 2021). In their original presentation, Hyperbolic networks++ omitted any activation because of the inherent non-linearity of the hyperbolic space. To ensure a fair comparison, we added ReLU activations between layers, as well as an initial embedding into the Poincaré manifold through the exponential map.

### I.3 NUMBER OF OPERATIONS

A single linear Cartan layer transforming a batch of $B$ input points of dimension $D$ into an output of dimension $M$ has an (rough) estimated floating-point operation (FLOP) count of

$$\text{FLOPs} \approx \underbrace{B(2(D-1))(M-1)}_{\text{matrix mult.}} + \underbrace{B(M-1)}_{\text{bias}} + \underbrace{2BM + 19B}_{\text{group op.}} + \underbrace{2B(3M-4) + 53B}_{\text{rotation}}$$

$$= 2BDM - 2BD + 7BM + 65B,$$

where we counted $\approx 20$ FLOPs for each logarithm/ exponential operation. In comparison, a normal linear Euclidean layer has $\approx 2BDM + BM$ FLOPs. While this is a very crude approximation of the number of operations involved in our model, it gives an initial estimate of the difference between hyperbolic and Euclidean layers.

### I.4 EXPERIMENTAL HYPERPARAMETERS

Experimental hyperparameters for the numerical simulations are detailed in Tables S1-S2. For the AlexNet experiment, the fiber rotation parameter is high-dimensional and thus very sensitive during gradient descent; hence, we divided its individual learning rate by a factor of 100.

Table S1: Experimental hyperparameters

| Problem | Optimizer | Loss | Activation | Scheduler lr |
|---|---|---|---|---|
| Fully Connected | SGD | Cross-entropy | ReLU | no |
| AlexNet | SGD | Cross-entropy | ReLU | Plateau |
| ResNet | SGD | Cross-entropy | ReLU | Plateau |

Due to the computationally intensive nature of the problem, classification datasets were optimized using early stopping with a buffer of 15 on the test loss for up to 1000 epochs, while the regression tasks were optimized for 5000 epochs.

Table S2: Experimental hyperparameters (continued)

| Problem | Learning rate | Weight decay |
|---|---|---|
| Fully Connected | $1.00 \times 10^{-2}$ | $1.00 \times 10^{-5}$ |
| AlexNet | $1.00 \times 10^{-2}$ | $5.00 \times 10^{-5}$ |
| ResNet | $1.00 \times 10^{-2}$ | $5.00 \times 10^{-5}$ |

## I.5 COMPUTATION TIME

Simulations reported in Table 1 were conducted with an NVIDIA Tesla T4 GPU, operating on a PCI-E Gen3 x16 slot. Total computation time for all runs of each model is reported below.

Table S3: Total computation time for fully connected networks

| Model | Total computation time |
|---|---|
| Euclidean | 9h 26m 28s |
| Hyperbolic++ | 18h 1m 31s |
| Cartan | 19h 58m 22s |
| Lorentz | 21h 14m 55s |
| Poincaré | 23h 25m 54s |

Table S4: Total computation time for experiments on convolutional networks

| Model | Total computation time |
|---|---|
| Alexnet | 43h 21m 01s |
| H-Alexnet | 69h 24m 27s |
| ResNet18 | 52h 24m 55s |
| H-ResNet18 | 101h 55m 02s |

