# OpenReview forum: "Cartan Networks: Group theoretical Hyperbolic Deep Learning"
_ICLR.cc/2026/Conference — Submitted to ICLR 2026_

### Official Review · Reviewer_xxbm · 2025-10-21

**Soundness:** 2
**Presentation:** 1
**Contribution:** 2
**Rating:** 4
**Confidence:** 2

**Summary:**

The paper proposes Cartan Networks, a hyperbolic deep learning architecture that treats hyperbolic space as a solvable Lie group (via its symmetric-space structure) and builds layers by alternating group homomorphisms and isometries. This yields closed-form “linear” layers (mixing Cartan/fiber coordinates) and a hyperbolic logistic/softmax based on geodesic hyperplanes.

**Strengths:**

1. Clear geometric formulation of isometries and homomorphisms with explicit coordinate formulas
2. The architecture is easy to implement with standard Riemannian optimizers (GeoOpt) and the code is promised.

**Weaknesses:**

1. Most of what is framed as new (stacking “linear” layers built from isometries + homomorphisms; softmax via hyperbolic hyperplanes) is already implicit or explicit in prior gyrovector/Poincaré constructions and Lie-group views of $H^n$. The paper itself notes that prior hyperbolic layers parametrize all isometries (via exp/log + parallel transport) and that its layer differs mainly by coordinate choice and homomorphism tying. This reads as a repackaging rather than a fundamentally new capability.
2.  Benchmarks are small, with parity on CelebA/CIFAR-10 and a single strong gain on CIFAR-100; there is no task where Euclidean/Poincaré fails but Cartan succeeds, nor graphs/hierarchies where hyperbolic geometry is a priori motivated. This makes the practical utility unclear.
3. Hyperbolic hyperplanes and distance-monotone logits are standard (e.g., Poincaré SVM/softmax). The paper acknowledges they are the same subspaces as prior work; the novelty is just the solvable-coord formula.

**Questions:**

1. What is truly new? Prior work already realizes isometries via exp/log pipelines, but the paper should prove that this yields strictly larger (or different) function classes than existing hyperbolic “linears,” beyond coordinate convenience.

---

> ### Author Response · Authors · 2025-11-19
>
> We would like to thank the reviewer for their insights into our work. We address each comment below in detail.
>
> > Q1. What is truly new? Prior work already realizes isometries via exp/log pipelines, but the paper should prove that this yields strictly larger (or different) function classes than existing hyperbolic “linears,” beyond coordinate convenience.
>
> > W1. Most of what is framed as new (stacking “linear” layers built from isometries + homomorphisms; softmax via hyperbolic hyperplanes) is already implicit or explicit in prior gyrovector/Poincaré constructions and Lie-group views of $H^n$. The paper itself notes that prior hyperbolic layers parametrize all isometries (via exp/log + parallel transport) and that its layer differs mainly by coordinate choice and homomorphism tying. This reads as a repackaging rather than a fundamentally new capability.
>
> > W3. Hyperbolic hyperplanes and distance-monotone logits are standard (e.g., Poincaré SVM/softmax). The paper acknowledges they are the same subspaces as prior work; the novelty is just the solvable-coord formula.
>
> We address all these comments together, because they reference the same issue. We recognize that certain aspects of our layer design are features already discussed in the relevant literature, which we have extensively referenced and cited. However, what is entirely novel in our work is the use of the metric equivalence between hyperbolic space and a suitable solvable Lie group. This result allows for the definition of homomorphisms between layers that preserve the underlying group structure. In this framework, the homomorphisms are not merely “tying” the architecture together, but are rather its conceptual foundation and primary innovation. As the reviewer rightly notes that this point was not sufficiently emphasized; we will revise specific sections of the manuscript to clarify it more effectively.
>
> Regarding the difference between previous hyperbolic function classes and ours, we remark that while all isometries can be realized as metric preserving morphisms of tangent spaces, the space of homomorphisms and the space of linear maps between tangent spaces are not necessarily overlapping. We conceptualize previous versions of the hyperbolic linear layers as
>
> $$\mathrm{Exp}_b \circ W \circ \mathrm{Log}_0$$
>
> where with $\mathrm{Exp_p}$ and $\mathrm{Log_p}$ we denote the Riemannian exponential and logarithmic map, respectively. While $\mathrm{Log_0}$ maps the manifold to its tangent space, it is not $\mathrm{Log_{0, \mathrm{Lie}}}$, the logarithmic map in the Lie group sense, which is possible since the metric on this space is not bi-invariant. Moreover, the relationship between the two maps is nonlinear, i.e. there is not matrix $A$ such that $\;A \circ \mathrm{Log_0} = \mathrm{Log_{0, \mathrm{Lie}}}$. Hence $W \circ \mathrm{Log_{0, \mathrm{Lie}}}$ does not trivially fall into the class of Lie algebra morphisms, as otherwise $W \circ \mathrm{Log_0} = W' \circ  \mathrm{Log_{0, \mathrm{Lie}}} \implies (W')^{-1}W \circ \mathrm{Log_0} = \mathrm{Log_{0, \mathrm{Lie}}}$.
>
> From this, it follows that we are parametrizing a strictly different set of functions compared to traditional HNN architectures.
>
> > W2. Benchmarks are small, with parity on CelebA/CIFAR-10 and a single strong gain on CIFAR-100; there is no task where Euclidean/Poincaré fails but Cartan succeeds, nor graphs/hierarchies where hyperbolic geometry is a priori motivated. This makes the practical utility unclear.
>
> As correctly noted, our experiments are limited to relatively simple datasets. However, we would like to clarify that the primary aim of this work is **to introduce Cartan networks as an alternative framework for hyperbolic deep learning**; naturally, the new framework is not (yet) optimised for scalability and accuracy, and comparisons with highly engineered architectures can be misleading. We focus on a new, theory-driven geometric framework for representation learning in non-Euclidean spaces. In this sense, our contribution is foundational and exploratory, similar to much of the early work on hyperbolic neural networks, which also focused on small-scale experiments. We believe that our results show the promise of the new approach, but we recognise that more extensive validation would be helpful to also recognise the relative merits against other hyperbolic architectures.

---

### Official Review · Reviewer_ipsL · 2025-10-28

**Soundness:** 2
**Presentation:** 2
**Contribution:** 2
**Rating:** 4
**Confidence:** 1

**Summary:**

This paper introduces a new type of deep learning architecture named Cartan networks, where hidden features lie on solvable Lie groups and where each layer consists of a homomorphism from the source to the target group followed by an isometry within the target group. This type of network is motivated by a connection between non-compact symmetric spaces (such as hyperbolic space) and solvable Lie groups. The new type of layer is tested in small networks on small datasets.

**Strengths:**

- The text of the paper seems well written.
- If the new formulation is correct, then it could be an interesting new perspective on hyperbolic learning.
- Again, if the formulation is correct, then automatically being able to generalize this kind of learning to any non-compact symmetric space could be very interesting as well.

**Weaknesses:**

I want to start by saying that I do not have a sufficiently strong background in Lie group theory for understanding the theoretical part of this paper, which spans most of it. As a result, I am incapable of providing any meaningful review of the theoretical part of the text.

I do have significant experience with hyperbolic representation learning and computer vision, so I believe that I can judge the experimental part of this paper. There are, in my opinion, a few weaknesses to point out in the experiments:
1. The clearest weakness is the choice of datasets. Currently, the experiments involve a toy dataset, MNIST (and variants), CIFAR-10/100 and CelebA. All of these datasets are considered far too small and simple for drawing any conclusion regarding large scale networks. Therefore, I think some larger experiments (on ImageNet for example) should be included to support the authors' findings.
2. The experiments use small fully connected networks and an AlexNet-like network architecture. In order to see whether this type of layer actually scales to deep neural networks, the experiments should at least include something like a ResNet architecture in my opinion.
3. The layer is compared to the HNN layer in the Poincaré ball model by (Ganea et al., 2018) and to the standard Euclidean layer. However, as mentioned by the authors in the previous literature section, (Shimizu et al., 2021) have proposed an updated variant of the Poincaré layer and (Chen et al., 2022) have also proposed a (geometrically different) hyperboloid layer. A proper comparison should include at least both of these networks. ResNet-like architectures involving these layers already exist (Bdeir et al., 2024, [1]) and could easily be compared against as well.
4. The authors mention an increase in computational complexity, but the experiments section does not seem to contain a clear comparison in terms of actual computation time. Such an experiment would be very insightful.
5. The results in the currently included experiments appear mixed to me, making it unclear to me what the benefit of this new formulation is.

Based on these weaknesses I am leaning towards rejection. However, given my lack of understanding of the theoretical parts, my rating should be taken with a grain of salt.

If the authors can add the experiments that I mentioned above and find strong results, then I would be willing to increase my rating.

[1] Max van Spengler, Erwin Berkhout, and Pascal Mettes. *Poincaré resnet.* Proceedings of the IEEE/CVF International Conference on Computer Vision, 2023.

**Questions:**

1. You mention in the discussion that this new architecture could be more interpretable than existing architectures. Can you explain this claim a bit further?
2. What does metric equivalence mean within the context of your paper? The only common definition that I know is when two metrics on the same set induce some equivalent structure such as the same topology. However, I'm not sure I understand how this would work when there are two different underlying spaces.

---

> ### Author Response · Authors · 2025-11-19
>
> We thank the reviewer for their comments.
>
> > Q1. You mention in the discussion that this new architecture could be more interpretable than existing architectures. Can you explain this claim a bit further?
>
> Since a similar question was raised by reviewer vSig, we provide the same response here. Regarding representations, we argue that hyperbolic layers are more interpretable than Euclidean ones, especially in our Paint group formulation, where the Cartan coordinate controls the non-linearity of the representation. For example, we initialize weights such that all points lie in the subspace $\tilde E =\{\Upsilon_1 = 0\}$ at initialization. In this subspace, the distance between points is $d_{ij}=\text{acosh}(1+\frac{1}{2}||\Upsilon_2^i- \Upsilon_2^j||^2)$, and the group operation is the "Euclidean" translation in the Paint coordinates. As discussed in the paper, with this initialization, the network starts as equivalent to a Euclidean network (with one fewer neuron in each layer), and it can then use the nonlinear parameters as needed. The distance of a generic point $\Upsilon$ from $\tilde E$ is given by $d(\Upsilon, \tilde E) = \Upsilon_1^2$, so this can be used as a measure of how "nonlinear" its representation is inside the network. In our datasets, we observed a weak correlation between data complexity and its nonlinearity measured this way.
>
> The previous paragraph was just a sketch of a possible way to use the hyperbolic structure, and should not be taken as final. The important conceptual point is that non-Euclidean spaces with structured geometric or topological properties might improve neural network interpretability, as these spaces impose known symmetries and relational constraints that make learned representations more transparent.
>
> > Q2. What does metric equivalence mean within the context of your paper? The only common definition that I know is when two metrics on the same set induce some equivalent structure, such as the same topology. However, I'm not sure I understand how this would work when there are two different underlying spaces.
>
> A Riemannian manifold $(\mathcal{M},g)$ is metrically equivalent to an Alekseevskian normal solvable group $\mathcal{S}$ if
> - $\mathcal{S}$ has a free transitive action on $\mathcal{M}$ that is an isometry for $g$. In this way, $\mathcal{M}$ is diffeomorphic to $\mathcal{S}$.
> - There exists on the solvable Lie algebra $Solv$ of the solvable group $\mathcal{S}$ a positive definite symmetric quadratic form $\langle \:, \: \rangle$ whose $\mathcal{S}$ transport from the origin to any point of $\mathcal{M}\sim \mathcal{S}$ coincides with
> the metric $g$.
>
> In simpler terms, the metric equivalence between a manifold and a group means that:
>  - the group acts on the manifold via isometries,
>  - there is a metric-preserving map between the manifold and the group equipped with a left-invariant metric.
>
> > W1. The clearest weakness is the choice of datasets. Currently, the experiments involve a toy dataset, MNIST (and variants), CIFAR-10/100, and CelebA. All of these datasets are considered far too small and simple for drawing any conclusion regarding large-scale networks. Therefore, I think some larger experiments (on ImageNet, for example) should be included to support the authors' findings.
>
> > W2. The experiments use small fully connected networks and an AlexNet-like network architecture. In order to see whether this type of layer actually scales to deep neural networks, the experiments should at least include something like a ResNet architecture in my opinion.
>
> We tackle TinyImageNet, a scaled-down version of ImageNet with 200 classes out of 1000, using both our hyperbolic version of AlexNet, as described in the manuscript, and a hyperbolic version of ResNet18, obtained by generalizing Euclidean layers with the same technique. The results are shown below:
>
>
> | Problem     | AlexNet         | H-AlexNet      | ResNet18 | H-ResNet18 |
> | ------------- | ---------------- | ---------------- | -------------- | ---------------------------- |
> | TinyImageNet | 0.446 ± 0.003 | 0.544 ± 0.003 | 0.555 ± 0.006 | 0.544 ± 0.005|
>
> As we can see, we obtain a significant boost in performance on TinyImagenet for fixed experimental configurations with AlexNet, while the performance of the ResNet is comparable. Preliminary data suggests that there is a strong dependency on the learning rate for AlexNet and its hyperbolic variant, so we are working on a hyperparameter search together with a ResNet comparison with Bdeir et al., 2024.

---

> ### Author Response · Authors · 2025-11-19
>
> > W3. The layer is compared to the HNN layer in the Poincaré ball model by (Ganea et al., 2018) and to the standard Euclidean layer. However, as mentioned by the authors in the previous literature section, (Shimizu et al., 2021) have proposed an updated variant of the Poincaré layer and (Chen et al., 2022) have also proposed a (geometrically different) hyperboloid layer. A proper comparison should include at least both of these networks. ResNet-like architectures involving these layers already exist (Bdeir et al., 2024, [1]) and could easily be compared against as well.
>
> Following your suggestion and that of other reviewers, we expanded the results on fully connected architectures to test on *Fully Hyperbolic* and *Hyperbolic++* architectures. The articles presenting those architectures have a strong focus on transformer architectures, which are not considered in our work. We then had to make a few adjustments to adapt those layers to this problem. This comparison should be taken as exploratory, because an in-depth review of the performance of existing hyperbolic architectures was beyond the scope of our work. Here, we have used ReLU as activation, fixed lr = 0.01, and RSGD as optimizer, and tested across different numbers of layers/neurons. We highlight the model with the best performance (or the performances that are statistically tied).
>
> | Problem     | Cartan         | Euclidean      | Hyperbolic++ | Fully hyperbolic | Poincaré     |
> | ------------- | ---------------- | ---------------- | -------------- | ---------------------------- | -------------- |
> | Cifar10 | 52.6 ± 0.3   | 52.6 ± 0.5   |52.4 ± 1|  52.4 ± 0.8  | 52.5 ± 0.3 |
> | FMNIST  | **89.3 ± 0.3**   | **89.3 ± 0.1**   | 87.9 ± 0.5     | **89.2 ± 0.2**               | **89.4 ± 0.2** |
> | KMNIST  | 90.10 ± 0.07 | 90.0 ± 0.1       | **89.0 ± 0.1** | **90.29 ± 0.1**             | **90.2 ± 0.2** |
> | MNIST   | **98.27 ± 0.02** | **98.27 ± 0.02** | 98.0 ± 0.1     | 98.14 ± 0.04                  | 98.19 ± 0.6    |
>
> We are currently in the process of expanding the results on Resnet to include a *Fully hyperbolic* architecture.
>
> > W4. The authors mention an increase in computational complexity, but the experiments section does not seem to contain a clear comparison in terms of actual computation time. Such an experiment would be very insightful.
>
> The total computational time for the experiments discussed in the previous question is summarized below. All hyperbolic models display comparable computational costs.
>
> | *Type*           | *Total Time (h:m:s)* |
> | ---------------- | ----------------------- |
> | Euclidean        | 9:26:28                 |
> | Hyperbolic++     | 18:01:31                |
> | Cartan           | 19:58:22                |
> | Fully hyperbolic | 21:14:55                |
> | Poincaré         | 23:25:54                |
>
> The total computational time for the experiments provided on Alexnet and its hyperbolic version is
>
> | *Type*           | *Total Time (h:m:s)*    | Machine |
> | ---------------- | ----------------------- | --------|
> | Alexnet          | 43:21:01                | Tesla P100/Xeon E5-2683 v4 |
> | H-Alexnet        | 69:24:27                | Tesla P100/Xeon E5-2683 v4 |
>
> while for the TinyImagenet experiment on the Resnet we have
>
> | *Type*           | *Total Time (h:m:s)*    | Machine |
> | ---------------- | ----------------------- | ------- |
> | Resnet          | 19:55:02                | Tesla P100/Xeon E5-2683 v4 |
> | H-ResNet18       | 16:24:55                | Tesla A100/Intel(R) Xeon(R) Gold 6238R CPU @ 2.20GHz |
>
>
> > W5. The results in the currently included experiments appear mixed to me, making it unclear to me what the benefit of this new formulation is.
>
>
> We agree that the improvements over the baseline are not sharp, as they are in line with previous implementations showing at best moderate accuracy enhancements, as, for example, in Bdeir et al. (2024), which was motivated by the exploration of a hyperbolic version of the convolutional networks. The main objective of the work is not to introduce a new best-performing architecture, but rather to explore a different, geometrically motivated way of building neural networks, with benefits in terms of interpretability and ease of implementation.
>
> The solvable group structure is interesting in general, as it allows one to distinguish coordinates at the various levels of the derived series. In our case, which is the simplest group in our considered manifold class, the graded structure allows us to separate the Cartan and other coordinates, the latter being a parametrization of Euclidean space, and the former a hyperbolic addition. This is precisely what allows a flexible function class that extends Euclidean architectures, allowing us to generalize (with some choices) previously established models like ResNet and AlexNet.

---

> > ### Comment · Reviewer_ipsL · 2025-11-26
> >
> > I thank the authors for their responses and for adding the additional results. Based on the latter I agree with the authors that the proposed architecture does not show any clear improvements over existing hyperbolic architectures with respect to performance or computational efficiency. Therefore, as the authors state, I too think its potential value should be based on any merits it may have in terms of interpretability or ease of use.
> >
> > The authors claim that their proposed architecture improves both of these. However, this is not obvious to me and this is what constitutes my primary concern with the paper. I will try to explain my concerns:
> > 1. In commonly used models of hyperbolic space (e.g. the Poincaré ball), points can be represented as vectors with a clear physical meaning that seems to me quite interpretable. In fact, these models typically allow visualization of $\mathbb{H}^2$ and $\mathbb{H}^3$ in a similar fashion as we visualize Euclidean space, albeit with some distortion of angles and distances. The representation used in the paper, where elements are parameterized matrices seems significantly more difficult to interpret and particularly to visualize (in a meaningful way) to me.
> > 2. This seems to be confirmed by Figure 2, where the authors have chosen the Poincaré ball for most of the visualization and where the effect of these operations is also quite clearly visible in that model, whereas the visualization in solvable coordinates is less clear to me. Moreover, the operations shown in subfigures b) and c) can easily be represented interpretably as Möbius transformations.
> > 3. If I understand correctly, then the linear layer involves an isometry on the space, which is "parameterized in terms of the Poincaré ball coordinates". If that is the case, then what is the benefit of using the proposed representation instead of simply opting for the Poincaré ball model? It seems to me that this would indeed heavily simplify the fiber rotation given in equation (11).
> > 4. It is claimed that one of the main benefits of the proposed linear layer lies in the fact that it does not break symmetries between layers whereas existing architectures (such as HNN and HNN++) do. This is based on those architectures following equation (14). This is false for HNN++ (Shimizu et al., 2021). Their definition is based on the distance-to-hyperplane interpretation. There are a few points here that I don't fully understand: 1) what is meant with symmetry breaking; 2) why is it a problem; and 3) does HNN++ suffer from the same problem?
> > 5. I don't understand why the proposed architecture is easier to implement. The other existing hyperbolic architectures such as HNN, HNN++ and the Lorentz version (Chen et al., 2022) are very easy to implement in various architectures as well.  The argument in lines 360-369 uses convolutions as an example, but building convolutional models out of the previously existing architectures is very straightforward, as shown for instance in (Bdeir et al., 2024) and [1].
> >
> > [1] Max van Spengler, Erwin Berkhout, and Pascal Mettes. Poincaré resnet. Proceedings of the IEEE/CVF International Conference on Computer Vision, 2023.

---

> > > ### Author Response · Authors · 2025-11-27
> > >
> > > Firstly, we would like to respond by saying that, while we agree that considerations of performance, ease of use, and interpretability played a role in our thinking about these architectures, the main motivation behind our work is theoretical. In this work, we propose and explore a different perspective, that of the isometry with a solvable group, and discuss how this can guide model design.
> > >
> > > > P1 In commonly used models of hyperbolic space (e.g., the Poincaré ball), points can be represented as vectors with a clear physical meaning that seems to me quite interpretable. In fact, these models typically allow visualization of and
> > > in a similar fashion as we visualize Euclidean space, albeit with some distortion of angles and distances. The representation used in the paper, where elements are parameterized matrices, seems significantly more difficult to interpret and particularly to visualize (in a meaningful way) to me.
> > >
> > > Thank you for the question. Our choice of coordinate system is not meant to introduce an alternative geometric interpretation of hyperbolic space. We chose these coordinates because they make the numerical implementation of the transformations used in our model easier and more stable: the architecture itself does not depend on this particular coordinate choice. All layers in the paper can be rewritten in the Poincaré model, the Lorentz model, or any other standard coordinate system for hyperbolic space. Since the change of coordinates between these models is fully analytical, it is straightforward to switch from one system to another depending on the task. In a way, this is similar to the choice between polar and Cartesian coordinates. Similarly to polar coordinates, we single out one component with a different meaning, the "Cartan" $\Upsilon_1$.
> > >
> > > Notice also that, while the solvable coordinates indeed parametrize matrices, in practice, all operations are always written in terms of the coordinates, and one never has to use the matrix representation.
> > >
> > > For visualization purposes, especially when illustrating isometries, we find Poincaré coordinates more intuitive and interpretable. However, for parameterizing the specific homomorphisms we work with, solvable coordinates are more convenient. If one prefers to work in the Poincaré model, they can convert all learned representations to Poincaré coordinates after training without any loss of information. The choice of coordinates is therefore a matter of numerical convenience rather than a conceptual requirement of the architecture. See, for example, Mishne et al., 2023, for a discussion of problems with the numerical stability of Poincarè coordinates.
> > >
> > > > P2 This seems to be confirmed by Figure 2, where the authors have chosen the Poincaré ball for most of the visualization and where the effect of these operations is also quite clearly visible in that model, whereas the visualization in solvable coordinates is less clear to me. Moreover, the operations shown in subfigures b) and c) can easily be represented interpretably as Möbius transformations.
> > >
> > > The operations that can be understood as Möbius transformations are the set of isometries, which has been one of the bases of hyperbolic learning. For example, in lines 252-266, we explicitly show how to find the set of isometries in terms of the Riemannian exponential and logarithmic maps.
> > >
> > > A key contribution of our work is to emphasize the metric equivalence between hyperbolic space and a Lie group, which allows us to introduce homomorphisms as additional, well-structured transformations. To the best of our knowledge, this class of transformations has not yet been explored for model design in hyperbolic deep learning. This tie with Lie Group theory could allow extensions beyond the hyperbolic setting to other symmetric spaces, where these operations are still well defined, whereas Möbius transformations are specific to hyperbolic geometry.
> > >
> > > > P3 If I understand correctly, then the linear layer involves an isometry on the space, which is "parameterized in terms of the Poincaré ball coordinates". If that is the case, then what is the benefit of using the proposed representation instead of simply opting for the Poincaré ball model? It seems to me that this would indeed heavily simplify the fiber rotation given in equation (11).
> > >
> > > In our model, all the transformations are parametrized in terms of the solvable coordinates. At lines 203-204, we were just pointing out that the isometries were already parameterized with Poincarè coordinates by previous works in the context of hyperbolic deep learning. In general, it is indeed correct that isometries might be easier to parameterize in the Poincarè coordinates. Homomorphisms, however, are quite simple in solvable coordinates, and become much more complicated in other Coordinate systems. There is indeed a trade-off between the choice of coordinates, which only impacts computation costs and numerical stability, but not the functional class one wishes to represent.

---

> > > > ### Author Response · Authors · 2025-11-27
> > > >
> > > > > P4 It is claimed that one of the main benefits of the proposed linear layer lies in the fact that it does not break symmetries between layers, whereas existing architectures (such as HNN and HNN++) do. This is based on those architectures following equation (14). This is false for HNN++ (Shimizu et al., 2021). Their definition is based on the distance-to-hyperplane interpretation. There are a few points here that I don't fully understand: 1) what is meant by symmetry breaking; 2) why is it a problem; and 3) does HNN++ suffer from the same problem?
> > > >
> > > > We agree with the reviewer that the claims that previous architectures broke the symmetry might have been imprecise. With lines 262-266, we meant to say that "the map induced by W is not covariant with respect to the isometry group". Concretely, we mean that interleaving homomorphisms with isometries extends the function class that we are parametrizing. If homomorphisms were covariant, then the role of isometries would be diminished, and our architecture (after isometric reparametrization of each layer) would be equivalent to one with homomorphisms only. We will emend the manuscript accordingly.
> > > >
> > > > > P5 I don't understand why the proposed architecture is easier to implement. The other existing hyperbolic architectures such as HNN, HNN++ and the Lorentz version (Chen et al., 2022) are very easy to implement in various architectures as well. The argument in lines 360-369 uses convolutions as an example, but building convolutional models out of the previously existing architectures is very straightforward, as shown for instance in (Bdeir et al., 2024) and [1].
> > > >
> > > > We fully agree, and our intention in lines 360–369 is not to argue that our architecture is easier to implement.  That paragraph shows how our architecture directly generalizes normally understood Euclidean layers to hyperbolic geometry, in the sense that our models can find a solution that is comparable to that found by an Euclidean model with one fewer neuron. If in other parts of the manuscript suggest claims about improved ease of implementation, we would like the reviewer to point it out, so that we can discuss it or clarify the wording.
> > > >
> > > > Our previous rebuttal claimed generic "benefits in terms of [...] ease of implementation" in the sense that our proposed framework suggests a straightforward generalization of Euclidean layers; we did not attempt to compare the actual implementations of previous hyperbolic architectures, which, given the various strategies that are employed in the literature to reproduce known layers, would be very hard to quantify.
> > > >
> > > > We would like to add a brief comparison of computation costs, to remark that straightforward implementations might come with a price. Bdeir et al. (2024) rely on unfolding convolutional operators, making the GPU and computational requirements heavily dependent on the input size. For comparison, we were not able to train on reasonable batches (32) of 224x224 images on their architectures on 40GB of GPU, while our architecture comfortably sits in the range of 20GB, (approx 10GB for traditional ResNet18).

---

> > > > > ### Author Response · Authors · 2025-12-03
> > > > >
> > > > > We additionally performed a lr parameter sweep on the ResNet, with these results.
> > > > >
> > > > > | *Type*     | *Total Time (h:m:s)* | Accuracy   |
> > > > > | ---------- | -------------------- | ---------- |
> > > > > | Resnet18   | 52:24:55             | 61.4 ± 0.2 |
> > > > > | H-Resnet18 | 101:55:02            | 61.5 ± 0.1 |
> > > > >
> > > > > We attempted testing the architecture against the resnet proposed in Bdeir et al., but were unable since their proposed implementation involves unfolding convolutional operators and hence has excessive memory requirements.

---

### Official Review · Reviewer_53Yr · 2025-10-31

**Soundness:** 3
**Presentation:** 3
**Contribution:** 2
**Rating:** 2
**Confidence:** 4

**Summary:**

This paper proposes “Cartan networks,” a hyperbolic deep learning architecture that treats each layer as a solvable Lie group and maps between layers as a composition of a group homomorphism and an isometry. They proposed linear layers as a composition of homomorphisms between solvable groups with associated softmax and regression layers. They showed that adding pointwise activations recovers universal approximation and contains Euclidean networks as a special case. Their benchmark on synthetic and image classification tasks show in general comparable performance to Euclidean and tangent-space-based Poincare models.

**Strengths:**

1. The proposed hyperbolic linear and regression layers based on group isomorphisms/homorphisms and series of solvable groups is interesting
2. The proposed layers are defined by operations that are rather intrinsic, which is a potential useful way to improve hyperbolic neural network operations
3. Experiments show incorporating activation in the model leads to improvement performance, which is consistent with the theory presented in the paper

**Weaknesses:**

1. The experiments show more or less comparable performance against the Euclidean and Poincare model, with the few cases of noticeable improvements being synthetic datasets. This shows relatively weak motivation for the need for the model and whether the proposed method is actually effective
2. The baseline Poincare model is rather naive and simple. Since Ganea et al., 2018, many works have proposed hyperbolic linear and activation layers that lead to significant improvements (e.g. Chen et al. 2022, Shimizu et al. 2021, Yang et al. 2024). Missing comparison with these methods weakens the claims about the effectiveness of the proposed method as well. Coincidently, Chen et al. 2022 is also falsely cited to use nonlinearity on the tangent space on lines 358-359, when in reality they incorporate the nonlinearity into a fully hyperbolic layer. Yang et al. 2024 also does not use tangent space for nonlinearity.

Yang et al. 2024: https://arxiv.org/abs/2407.01290

3. The proposed method incurs computational overhead and rigidity into the model, even against some of the newer hyperbolic methods, but no analysis or discussion is given for this aspect.

**Questions:**

1. Since the proposed method does not show convincing improvements over the baselines, especially against the Euclidean model on image tasks, why strong are the motivation for this method? Can the authors find an application or dataset where preserving the group structure of hyperbolic manifolds is actually beneficial?
2. Also see weakness

---

> ### Author Response · Authors · 2025-11-19
>
> We thank you for your thorough comments and questions, which we address in detail below.
>
> > Q1. Since the proposed method does not show convincing improvements over the baselines, especially against the Euclidean model on image tasks, why strong are the motivation for this method? Can the authors find an application or dataset where preserving the group structure of hyperbolic manifolds is actually beneficial?
>
> > W1. The experiments show more or less comparable performance against the Euclidean and Poincare model, with the few cases of noticeable improvements being synthetic datasets. This shows relatively weak motivation for the need for the model and whether the proposed method is actually effective
>
> We agree that the improvements over the baseline are not sharp, as they are in line with previous implementations showing at best moderate accuracy improvements, as, for example, in Bdeir et al. (2024), which was motivated by the exploration of a hyperbolic version of convolutional networks. The objective of our work is not to introduce a new best-performing architecture, but rather to explore a different, geometrically motivated way of building neural networks, with benefits in terms of interpretability and ease of implementation.
>
> The solvable group structure is interesting in general, as it allows one to distinguish coordinates at the various levels of the derived series. In our case, which is the simplest group in our considered manifold class, the graded structure allows us to separate the Cartan and other coordinates, the latter being a parametrization of Euclidean space, and the former a hyperbolic addition. This is precisely what allows a flexible function class that extends Euclidean architectures, allowing us to generalize (with some choices) previously established models like ResNet and AlexNet.
>
> > W2. The baseline Poincare model is rather naive and simple. Since Ganea et al., 2018, many works have proposed hyperbolic linear and activation layers that lead to significant improvements (e.g. Chen et al. 2022, Shimizu et al. 2021, Yang et al. 2024). Missing comparison with these methods weakens the claims about the effectiveness of the proposed method as well. Coincidently, Chen et al. 2022 is also falsely cited to use nonlinearity on the tangent space on lines 358-359, when in reality they incorporate the nonlinearity into a *Fully hyperbolic* layer. Yang et al. 2024 also does not use tangent space for nonlinearity.
>
> Regarding the incorrect citation, we thank you for pointing out this inconsistency, which we will fix in the manuscript. Regarding the general comparison with the Poincarè model, we have to stress that our main goal with this work was theoretical: demonstrating how the intrinsic metric equivalence with a Lie group could be used during network design to build new models. To this end, we argue that our architecture resembles more closely the approach of Ganea et al. Our architecture lacks the sophistication of later works because it was besides our intention.
>
> Still, we expanded the results on fully-connected architectures to test on *Fully hyperbolic* (Chen, 2022) and *Hyperbolic++* (Shimizu, 2021) architectures. These articles have a strong focus on transformer architectures, which are not considered in our work, so we adapted their implementation to our tasks. This comparison should be taken as exploratory, as an in-depth review of the performance of existing hyperbolic architectures was beyond the scope of our work. Here, we have taken ReLU as activation, fixed lr = 0.01, and RSGD as optimizer, and tested across different numbers of layers/neurons. We highlight the model with the best performance (or the performances that are statistically tied).
>
>
> | Problem     | Cartan         | Euclidean      | Hyperbolic++ | Fully hyperbolic | Poincaré     |
> | ------------- | ---------------- | ---------------- | -------------- | ---------------------------- | -------------- |
> | Cifar10 | 52.6 ± 0.3   | 52.6 ± 0.5   |52.4 ± 1|  52.4 ± 0.8  | 52.5 ± 0.3 |
> | FMNIST  | **89.3 ± 0.3**   | **89.3 ± 0.1**   | 87.9 ± 0.5     | **89.2 ± 0.2**               | **89.4 ± 0.2** |
> | KMNIST  | 90.10 ± 0.07 | 90.0 ± 0.1       | **89.0 ± 0.1** | **90.29 ± 0.1**             | **90.2 ± 0.2** |
> | MNIST   | **98.27 ± 0.02** | **98.27 ± 0.02** | 98.0 ± 0.1     | 98.14 ± 0.04                  | 98.19 ± 0.6    |
>
> We are currently working on expanding the section on Resnet architectures to include the *Fully hyperbolic* (Bdeir, 2024) model (see answer to reviewer vSig).

---

> > ### Author Response · Authors · 2025-11-19
> >
> > > W3. The proposed method incurs computational overhead and rigidity into the model, even against some of the newer hyperbolic methods, but no analysis or discussion is given for this aspect.
> >
> > The total computational time for the experiments discussed in the previous question is summarized below. All hyperbolic models display comparable computational costs.
> >
> > | *Type*           | *Total Time (h:m:s)* |
> > | ---------------- | ----------------------- |
> > | Euclidean        | 9:26:28                 |
> > | Hyperbolic++     | 18:01:31                |
> > | Cartan           | 19:58:22                |
> > | Fully hyperbolic | 21:14:55                |
> > | Poincaré         | 23:25:54                |
> >
> > The total computational time for the experiments provided on Alexnet and its hyperbolic version is
> >
> > | *Type*           | *Total Time (h:m:s)*    | Machine |
> > | ---------------- | ----------------------- | --------|
> > | Alexnet          | 43:21:01                | Tesla P100/Xeon E5-2683 v4 |
> > | H-Alexnet        | 69:24:27                | Tesla P100/Xeon E5-2683 v4 |
> >
> >
> >
> > while for the TinyImagenet experiment on the Resnet we have:
> >
> >
> > | *Type*           | *Total Time (h:m:s)*    | Machine |
> > | ---------------- | ----------------------- | ------- |
> > | Resnet          | 19:55:02                | Tesla P100/Xeon E5-2683 v4 |
> > | H-ResNet18       | 16:24:55                | Tesla A100/Intel(R) Xeon(R) Gold 6238R CPU @ 2.20GHz |
> >
> >
> > We refer to Appendix G.4 for a FLOPS estimate in comparison with Euclidean architectures.
> >
> > We are unsure about what the reviewer means by "rigidity" and would ask him to clarify the term.

---

### Official Review · Reviewer_vSig · 2025-11-01

**Soundness:** 3
**Presentation:** 3
**Contribution:** 3
**Rating:** 6
**Confidence:** 4

**Summary:**

This paper proposes Cartan networks which is a new class of hyperbolic deep learning algorithms where group homomorphisms are interleaved with metric-preserving diffeomorphisms. In particular, the authors highlight the metric equivalence of the hyperbolic space with a solvable Lie group to exploit the group structure as a tool in architectural design. Extensive experiments show the proposed architectures achieve competitive or better performance w.r.t. Euclidean and standard hyperbolic neural networks on real and synthetic datasets.

**Strengths:**

1. The proposed Cartan networks posses intrinsic Hyperbolic architecture which avoids the need for tangent-space approximations (as in Poincaré networks), making computations geometrically consistent and potentially more expressive.

2. Cartan networks allow stacking of homomorphisms and isometries, giving architectural flexibility and they are compatible with standard deep learning techniques (convolutions, batch normalization, dropout).

3. Cartan networks outperform Euclidean and standard hyperbolic networks on several benchmarks, including regression tasks, MNIST variants, CIFAR-100, and convolutional model.

**Weaknesses:**

1. The proposed architectures seem complex and require careful handling of solvable coordinates, fiber rotations, and Lie group operations, which may be non-trivial to implement. Also, it is not clear if the the proposed architecture can be applied to large datasets and extended to larger architectures.

2. Although parameter-efficient compared to some Euclidean layers, the matrix multiplications and exponential/fiber rotations could introduce extra computational cost.

3. The use of solvable group coordinates, fiber rotations, and Cartan/pain coordinates makes the learned representations even less interpretable compared to standard Euclidean embeddings.

**Questions:**

1. How do Cartan networks perform on very large-scale datasets or in high-dimensional hyperbolic spaces? Are there practical limits due to the complexity of solvable group operations and fiber rotations?

2. Can the representations learned by Cartan networks be transferred across tasks or datasets, similar to pre-trained Euclidean or graph neural networks? How well do they generalize beyond the training domain?

---

> ### Author Response · Authors · 2025-11-19
>
> We appreciate your feedback and comments! Below, we address the questions and weaknesses, grouping them when we feel that will help the discussion.
>
> > Q1. How do Cartan networks perform on very large-scale datasets or in high-dimensional hyperbolic spaces? Are there practical limits due to the complexity of solvable group operations and fiber rotations?
>
> > W1. The proposed architectures seem complex and require careful handling of solvable coordinates, fiber rotations, and Lie group operations, which may be non-trivial to implement. Also, it is not clear if the the proposed architecture can be applied to large datasets and extended to larger architectures.
>
> We agree that the addition of hyperbolic operations sometimes increases the numerical instability of the problem (and previous hyperbolic implementations are not strangers to this, as reported by https://arxiv.org/abs/2211.00181). We encountered problems in the fiber rotation operation when more complex architectures were involved. This was mostly solved by gradient clipping.
>
> To provide additional evidence on the scalability of the approach, we tackle TinyImageNet, a scaled-down version of ImageNet with 200 classes out of 1000, with both our hyperbolic version of Alexnet described in the manuscript and a hyperbolic version of ResNet18, obtained by generalizing Euclidean layers with the same technique. The results are shown below:
>
>
> | Problem     | AlexNet         | H-AlexNet      | ResNet18 | H-ResNet18 |
> | ------------- | ---------------- | ---------------- | -------------- | ---------------------------- |
> | TinyImageNet | 0.446 ± 0.003 | 0.544 ± 0.003 | 0.555 ± 0.006 | 0.544 ± 0.005|
>
> As we can see, we obtain a significant boost in performance on TinyImagenet for fixed experimental configurations with AlexNet, while the performance of the ResNet is comparable. Preliminary data suggests that there is a strong dependency on the learning rate for AlexNet and its hyperbolic variant, so we are working on a hyperparameter search together with a ResNet comparison with Bdeir et al., 2024.
>
> We believe the experimental data provided, together with the tests on CelebA and Cifar100 reported in the manuscript, provide good evidence that the approach is scalable.
>
> > W2. Although parameter-efficient compared to some Euclidean layers, the matrix multiplications and exponential/fiber rotations could introduce extra computational cost.
>
> We acknowledge that this is probably the main weakness of our work on this topic, and it is broadly shared by the entire field of Hyperbolic Neural Networks. We argue that the spirit of HNNs is to introduce complexity in the architecture in the hope that this will lead to networks performing better with fewer parameters on tasks where the hyperbolic structure plays a significant role. For a more detailed comparison between our and other hyperbolic architectures, we refer to our reply to reviewer 53Yr. To summarize, computation time for our networks aligns with commonly used hyperbolic neural networks. We refer to Appendix G.4 for a comparative FLOPS estimate.
>
> > Q2. Can the representations learned by Cartan networks be transferred across tasks or datasets, similar to pre-trained Euclidean or graph neural networks? How well do they generalize beyond the training domain?
>
> This is a relevant question and an interesting avenue for further exploration. We believe, given the similarity in performance and in function class with Euclidean networks, that our networks can generalize in a meaningful way at least as well as traditional architectures. However, we decided to focus on other experimental endeavors closer to the objectives of the manuscript for the scope of this discussion.

---

> > ### Author Response · Authors · 2025-11-19
> >
> > > W3. The use of solvable group coordinates, fiber rotations, and Cartan/pain coordinates makes the learned representations even less interpretable compared to standard Euclidean embeddings.
> >
> > Regarding representations, we argue that hyperbolic layers are more interpretable than Euclidean ones, especially in our Paint group formulation, where the Cartan coordinate controls the non-linearity of the representation. For example, we initialize weights such that all points lie in the subspace $\tilde E =\{\Upsilon_1 = 0\}$ at initialization. In this subspace, the distance between points is $d_{ij}=\text{acosh}(1+\frac{1}{2}||\Upsilon_2^i- \Upsilon_2^j||^2)$, and the group operation is the "Euclidean" translation in the Paint coordinates. As discussed in the paper, with this initialization, the network starts as equivalent to a Euclidean network (with one fewer neuron in each layer), and it can then use the nonlinear parameters as needed. The distance of a generic point $\Upsilon$ from $\tilde E$ is given by $d(\Upsilon, \tilde E) = \Upsilon_1^2$, so this can be used as a measure of how "nonlinear" its representation is inside the network. In our datasets, we observed a weak correlation between data complexity and its nonlinearity measured this way.
> >
> > The previous paragraph was just a sketch of a possible way to use the hyperbolic structure, and should not be taken as final. The important conceptual point is that non-Euclidean spaces with structured geometric or topological properties might improve neural network interpretability, as these spaces impose known symmetries and relational constraints that make learned representations more transparent.

---

> > > ### Author Response · Authors · 2025-12-03
> > >
> > > We additionally performed a lr parameter sweep on the ResNet, with these results.
> > >
> > > | *Type*     | *Total Time (h:m:s)* | Accuracy   |
> > > | ---------- | -------------------- | ---------- |
> > > | Resnet18   | 52:24:55             | 61.4 ± 0.2 |
> > > | H-Resnet18 | 101:55:02            | 61.5 ± 0.1 |
> > >
> > > We attempted testing the architecture against the resnet proposed in Bdeir et al., but were unable since their proposed implementation involves unfolding convolutional operators and hence has excessive memory requirements.

---

### Author Response · Authors · 2025-11-19

We sincerely thank all four reviewers for thoroughly inspecting our manuscript. We briefly summarize and discuss the most important points raised in the different reviews and the changes we made to the manuscript.

- Different reviewers felt that we could have expanded our testing to larger datasets and architectures. To this end, we added a Cartan version of a *Resnet18*, and we tested it on *Tiny Imagenet*. We are in the process of testing it on the previously discussed datasets.
- We introduce an exploratory comparison with other popular models in the hyperbolic literature mentioned by reviewers, namely *Fully hyperbolic* (Chen, 2022) and *Hyperbolic++* (Shimizu, 2021) architectures. Given that these architectures have a strong focus on transformer architectures, which are not considered in our work, we adapted their implementation to our tasks. We are also testing these models without activations, and we expect to have updated results by the end of this week.
- We added a section discussing the computation time of our model as well as a comparison between other hyperbolic architectures.
- We rectified the error in the citation reported by 53Yr.
- Given the added experimental section, we believed the results on regression on toy datasets were no longer relevant, and we removed them.
- The original manuscript mentioned a diffeomorphic activation function (DiLU), which made the comparisons more challenging. We rerun the relevant experiments using ReLUs.

---

### Meta-Review · Area_Chair_iWki · 2026-01-10

**Summary:**

This paper introduces a hyperbolic deep learning framework that treats hyperbolic space through its metric equivalence with a solvable Lie group. Architecturally, layers interleave group homomorphisms (to exploit the solvable structure) with isometries (metric-preserving maps), aiming to provide an intrinsic alternative to tangent-space-based constructions and to broaden the design space for hyperbolic architectures.

Here are the main concerns and how the rebuttal addressed them.

1) Practical value vs. mostly theoretical contribution.
Multiple reviewers noted that the empirical gains over Euclidean and standard hyperbolic baselines are often modest, making the motivation for a more complex construction unclear. The rebuttal was candid: the primary goal is a new theoretically motivated design perspective, not necessarily immediate accuracy wins, and it reframed claims accordingly.

2) Missing / weak baselines against modern hyperbolic layers.
A key objection was that comparisons to a simple Poincaré baseline were insufficient. The rebuttal added an exploratory comparison against Hyperbolic++ and Fully Hyperbolic variants on several small benchmarks, and corrected an erroneous citation flagged by reviewers. These additions address the “outdated baseline” criticism, though the comparison remains limited in scope and scale.

3) Scalability to larger architectures and datasets.
Reviewers asked for evidence beyond small FC nets and AlexNet-like models. The rebuttal added TinyImageNet experiments with H-AlexNet and H-ResNet18, plus a learning-rate sweep on ResNet18. Results suggest the approach can be trained at this scale, but improvements are mixed (clear gain on AlexNet, parity on ResNet), and the compute overhead becomes more visible.

4) Compute cost and implementation complexity.
Concerns about overhead and numerical stability were addressed with reported training-time comparisons across hyperbolic baselines, notes on instability in fiber rotations mitigated via gradient clipping, and a FLOPs pointer. The rebuttal argues costs are broadly comparable to other hyperbolic methods, but the added ResNet sweep also indicates substantially longer runtime for the hyperbolic variant under the reported setup.

5) Interpretability and “symmetry breaking” claims.
One reviewer challenged claims about interpretability and an imprecise statement about symmetry/covariance in prior architectures (notably HNN++). The authors acknowledged imprecision, clarified what was meant (covariance with respect to isometries / function-class extension), and committed to revise wording. The interpretability argument remains suggestive rather than demonstrated.

Overall, the submission offers a principled and potentially interesting geometric reframing of hyperbolic networks via solvable Lie groups, and the rebuttal improves baseline coverage and provides first evidence of scaling. However, the empirical case remains mixed, the strongest claims shift toward “foundational/exploratory,” and key advantages over existing, simpler hyperbolic constructions are not yet convincingly demonstrated at scale.

**Reviewer Concerns:**

Here are the main concerns and how the rebuttal addressed them.

1) Practical value vs. mostly theoretical contribution.
Multiple reviewers noted that the empirical gains over Euclidean and standard hyperbolic baselines are often modest, making the motivation for a more complex construction unclear. The rebuttal was candid: the primary goal is a new theoretically motivated design perspective, not necessarily immediate accuracy wins, and it reframed claims accordingly.

2) Missing / weak baselines against modern hyperbolic layers.
A key objection was that comparisons to a simple Poincaré baseline were insufficient. The rebuttal added an exploratory comparison against Hyperbolic++ and Fully Hyperbolic variants on several small benchmarks, and corrected an erroneous citation flagged by reviewers. These additions address the “outdated baseline” criticism, though the comparison remains limited in scope and scale.

3) Scalability to larger architectures and datasets.
Reviewers asked for evidence beyond small FC nets and AlexNet-like models. The rebuttal added TinyImageNet experiments with H-AlexNet and H-ResNet18, plus a learning-rate sweep on ResNet18. Results suggest the approach can be trained at this scale, but improvements are mixed (clear gain on AlexNet, parity on ResNet), and the compute overhead becomes more visible.

4) Compute cost and implementation complexity.
Concerns about overhead and numerical stability were addressed with reported training-time comparisons across hyperbolic baselines, notes on instability in fiber rotations mitigated via gradient clipping, and a FLOPs pointer. The rebuttal argues costs are broadly comparable to other hyperbolic methods, but the added ResNet sweep also indicates substantially longer runtime for the hyperbolic variant under the reported setup.

5) Interpretability and “symmetry breaking” claims.
One reviewer challenged claims about interpretability and an imprecise statement about symmetry/covariance in prior architectures (notably HNN++). The authors acknowledged imprecision, clarified what was meant (covariance with respect to isometries / function-class extension), and committed to revise wording. The interpretability argument remains suggestive rather than demonstrated.

**Reviewer Scores:**

vSig (6): likely 0
Already marginal accept; the added TinyImageNet and compute discussion answers their questions but does not clearly strengthen the “why this over alternatives” case.

53Yr (2): likely +1 to 3
The rebuttal fixed the citation error, added modern baselines (Hyperbolic++, Fully Hyperbolic), and provided compute-time tables. Still, the reviewer’s main complaint -- lack of clear advantage on meaningful tasks -- largely remains.

ipsL (4): likely 0 (possibly -1)
They explicitly state post-rebuttal that performance/efficiency are not clearly better, and shift the bar to interpretability/ease-of-use where they remain unconvinced.

xxbm (4): likely 0 to +1
The rebuttal addresses “what is new” more directly (homomorphisms via solvable-group equivalence) and adds broader comparisons, but the limited scale and modest gains likely keep this reviewer near borderline.

---

### Decision · Program_Chairs · 2026-01-26

Reject